# Neural Graph Generation from Graph Statistics

**Kiarash Zahirnia[1], Yaochen Hu[2], Mark Coates[3], Oliver Schulte[1]** [*]
[1]Simon Fraser University, [2]Huawei Noah's Ark Lab, [3]McGill University
kzahirni@sfu.ca, yaochen.hu@huawei.com, mark.coates@mcgill.ca, oschulte@sfu.ca

## Abstract

We describe a new setting for learning a deep graph generative model (GGM) from aggregate graph statistics, rather than from the graph adjacency matrix. Matching the statistics of observed training graphs is the main approach for learning traditional GGMs (e.g, BTER, Chung-Lu, and Erdos-Renyi models). Privacy researchers have proposed learning from graph statistics as a way to protect privacy. We develop an architecture for training a deep GGM to match statistics while preserving local differential privacy guarantees. Empirical evaluation on 8 datasets indicates that our deep GGM generates more realistic graphs than the traditional non-neural GGMs when both are learned from graph statistics only. We also compare our deep GGM trained on statistics only, to state-of-the-art deep GGMs that are trained on the entire adjacency matrix. The results show that graph statistics are often sufficient to build a competitive deep GGM that generates realistic graphs while protecting local privacy.

## 1 Introduction

Graph generative models (GGMs) have produced many insights into fundamental processes in domains including biology, engineering, and social sciences. Current deep GGMs are based on training data with complete adjacency matrices [6, 43, 88, 89]. This paper presents *GenStat*, a new deep GGM architecture for a setting where the available graph data are summarized by graph statistics, not a complete adjacency matrix. We refer to this setting as *statistics-based* graph generation. Previous work in network analysis has introduced several parametric non-neural models that support statistics-based graph generation, such as the Chung-Lu model [5], and the BTER model [67]. The parametric models tend to generate less realistic graphs due to lack of expressive power.

Our main motivation for generating graphs from statistics is privacy preservation. There is a direct tension between releasing real graphs to the research community and privacy concerns of the individual entities (graph nodes) [63, 85]. A promising proposal to address it is to release synthetic graphs that preserve the original graph properties while guaranteeing a user-specified level of privacy [20, 41]. Benchmarking Graph Neural Networks (GNNs) is a use-case of GGMs with a privacy guarantee. Synthetic graphs generated by GGMs enable GNN research without information leakage [87].

Statistics-based graph generation supports the challenging use case of decentralized graphs [86], where privacy concerns rule out collecting adjacency matrices in a central repository. Common examples are social graphs [58], e.g., users' connections through their phone contact lists, face-to-face interactions, sexual and friendship networks, or distributed social networks, e.g., Mastodon [58]. A solution is to have each entity perturb its data and send the perturbed data to a curator [86]. However, collecting raw data locally, such as an entity's neighbour list, requires heavy noise injection to satisfy privacy, and may result in a dense, distorted graph with low utility. An alternative, used widely in industry including Google, Apple, and Microsoft [11, 19, 71, 78], is to collect, from each entity's ego-graph, *local node-level graph statistics [24][Ch.2]*, with a guarantee of Local Differential

---

[*]Supported by NSERC Canada Discovery Grant R611341

Privacy (LDP) as a strong privacy measure [58]. The perturbed node-level statistics are aggregated to summarize important graph properties [58, 86, 91]. The idea is that coarser-grained information requires much less noise to satisfy LDP [58]. The final step is to generate representative synthetic graphs for public release. Previous works use a parametric statistics-based GGM [34, 35, 48, 58, 63]. Our *GenStat* system is a neural alternative that generates substantially more realistic graphs. Figure 1 summarizes the privacy-preserving workflow.

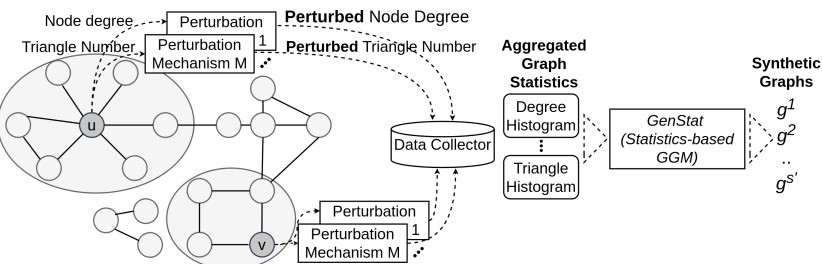

Figure 1: Generating realistic-looking graphs with Local Differential Privacy. See text for detail.

**Evaluation.** We evaluate the realism of the generated graph structures following previous work [6, 41, 58, 68, 72, 88]. Experiments on 8 datasets with diverse characteristics demonstrate the effectiveness of *GenStat*. Compared to parametric statistics-based methods, *GenStat* generates graphs that are up to 10 times more realistic on Maximum Mean Discrepancy (MMD) metrics. Compared to state-of-the-art GGMs (that require access to all adjacencies), the *GenStat* graphs reach very competitive graph quality, especially on real graph datasets, such as datasets of molecules and chemical compounds. Thus in many domains, graph statistics are sufficient for learning to generate realistic graphs. We also evaluate the effectiveness of *GenStat* for benchmarking GNNs on link prediction as a downstream task and we show *GenStat* is highly effective. Moreover, graph learning from statistics is much faster than from adjacency matrices. The implementation and datasets are provided at *GenStat* repository `https://github.com/kiarashza/GenStat.git` and explained in the Appendix Section 7.9.

**Contributions.**

- We introduce *GenStat*, a novel deep graph generative architecture that requires access only to graph-level statistics. To our knowledge, *GenStat* is the first deep GGM that does not require observations of individual nodes or edges.

- We show that given training statistics collected with a local differential privacy guarantee, *GenStat* also satisfies local differential privacy, while generating high-quality graphs.

- We identify permutation-invariant differentiable graph statistics that are based on aggregating node-level information and support realistic graph generation across multiple diverse domains.

## 2    Related work

**Realistic graph generation** has been studied extensively for decades, leading to the development of both parametric and neural approaches [24][Ch.8]. Figure 5 in the Appendix positions *GenStat* in the GGM landscape. The objective of previous parametric statistics-based GGMs such as [1, 5, 18, 21, 40, 56, 67] is to generate graphs with similar statistical properties to observed graphs [46]. The expressive power of their parameter space is limited. More recent machine learning models, including Deep GGMs [6, 88, 89], have a higher capacity to generate realistic graphs [6]. However, to our knowledge, they all assume access to a complete adjacency matrix. Such access is incompatible with privacy concerns since adjacency matrices reveal the entities with which an entity has interacted. In contrast, *GenStat* learns a deep GGM using only observed graph statistics and generates realistic graphs for a variety of domains.

**Graph statistics.** Zahirnia et al. [89] recently introduced a joint probabilistic micro-macro model over both adjacencies and graph statistics and showed that matching graph-level statistics is an effective regularizer for the edge reconstruction loss, which assumes access to all adjacencies. *GenStat* operates on permutation-invariant graph statistics that can be collected locally and privately, whereas some of the statistics used in [89] require all adjacencies.

**Statistical databases** are database systems that present only aggregate statistics (e.g., sample mean and count) for a subset of the entities represented in the database and ensure that sensitive information is safeguarded while still enabling valuable statistical analysis and research. An example is the database maintained by the U.S. Census Bureau [39]. Statistical databases also protect privacy through aggregate statistics and can be analyzed using the *GenStat* framework.

**Permutation-invariance** is a fundamental property of graph-structured data [24], which suggests that a GGM should produce the same output irrespective of the node ordering employed in the training set. It is satisfied by many GGMs [23, 28, 33, 54, 76], but not all [6, 88, 89]. Since *GenStat* takes as input permutation-invariant statistics, its training and the generative graph distribution are permutation-invariant as well (Section 3.3).

**Differential privacy** (DP) [15] enables the extraction of useful information about a population while providing strong privacy guarantees for individuals. *Local differential privacy (LDP)* [36] is a stronger guarantee for the decentralized setting [86]. Each entity perturbs its sensitive information before transmitting it to an untrusted curator. Collecting decentralized network statistics under LDP has been deployed by major technology companies, including Google, Apple, and Microsoft [11, 19, 71]. For example, Google's RAPPOR [19] collects randomized statistics to enable analysis of popular web destinations without revealing individual browsing habits.

**Graphs and privacy.** Privacy-preserving techniques have been proposed for the release and generation of graph data [41, 50, 51, 63]. Yang et al. [85] and Yoon et al. [87] leverage DP to enforce privacy constraints on deep GGMs. Yoon et al. [87] advocate using generated graphs to replace original graphs in GNN research. Therefore the GGM should generate effective benchmark graphs, meaning that GNNs show similar task performance as on the original source graphs. In the decentralized setting, entities perturb sensitive information locally to support the generation and release of synthetic graphs [58, 78, 86, 91]. Theoretical analysis shows that perturbing local adjacencies impairs graph quality too much for synthetic graphs to be useful [58]. Our experiments confirm this for SOTA deep GGMs (Section 4.3). Therefore, recent studies train parametric GGMs with perturbed local ego-graph statistics [58, 78] to achieve LDP. To our knowledge, *GenStat* is the first *deep GGM* that can generate synthetic graphs under LDP.

*Graph Anonymization* is a procedure that disguises or modifies information in graphs, making it anonymous. Anonymization techniques have two main limitations [50, 63]. These techniques 1) are mainly applicable in a centralized setting and 2) typically protect only against specific known attacks. Narayanan and Shmatikov [51] describe a de-anonymization algorithm for an anonymized binary adjacency matrix that effectively re-identifies the nodes in real word graphs, Twitter and Flicker.

*Federated Learning* is a privacy-preserving paradigm for building models from separate data sources [90]. [20, 25, 80] used federated learning to learn GNNs on graphs from multiple data sources. These studies assume that each source has trusted access to a sufficiently large subgraph to locally train an accurate GNN [80], and they do not target the construction of a GGM.

## 3 Problem definition and method

Given a set of observed graphs $\hat{G} = \{\mathcal{G}^1, ..., \mathcal{G}^S\}$, with variable number of nodes, sampled from a data distribution $p(G)$, the goal of GGMs is to learn a model that can generate similar synthetic graphs [6, 88, 89]. We introduce *GenStat*, a statistics-based architecture for probabilistic GGMs.

*GenStat* assumes that a training graph $\mathcal{G}^i$ is *summarized by a set of observed M graph statistics*, denoted as $\mathcal{I}^i = \{\mathcal{I}_m^i\}_{m=1}^M$, where the statistic $\mathcal{I}_m$ is a vector of dimension $D_m$. Each statistic is computed by a descriptor function $\phi_m : [0,1]^{n \times n} \to \mathbb{R}^{+D_m}$ that maps an adjacency matrix representing a graph with $n$ nodes to a vector; we write $\phi_m(\mathbf{A}^i) = \mathcal{I}_m^i$ and $\mathbf{\Phi}(\mathbf{A}^i) = \mathcal{I}^i$. Permutation-invariant statistics satisfy $\mathbf{\Phi}(\mathbf{A}^i) = \mathbf{\Phi}(\mathbf{A}_\pi^i)$ for all adjacency matrices obtained from $\mathbf{A}^i$ through a permutation $\pi$ [10, 52, Ch. 7]. Section 3.5 defines the statistics in our experiments, namely histograms of triangles, neighborhood sizes, random walks and graph size.

### 3.1 Probabilistic model of graph statistics

Given a distribution $p(\mathbf{A})$ over adjacency matrices, we let $\mathcal{I}$ denote the random variable defined by applying the descriptor functions to a random graph $\mathbf{A}$ (i.e., $\mathcal{I} = \{\phi_m(\mathbf{A})\}_{m=1}^M$). Following Ma et al.

[45], we view the latent adjacency matrix $\mathbf{A}$ as a sample from an underlying probabilistic adjacency matrix $\tilde{\mathbf{A}}$ with $\tilde{\mathbf{A}}_{u,v} \in [0,1]$ specifying independent link probabilities. We define the mixture model

$$p(\mathbf{A}|\tilde{\mathbf{A}}) = \prod_{u=1}^{n} \prod_{v=1}^{n} \tilde{\mathbf{A}}_{u,v}^{\mathbf{A}_{u,v}} (1 - \tilde{\mathbf{A}}_{u,v})^{1-\mathbf{A}_{u,v}} \tag{1}$$

$$p(\mathbf{A}) = \int p(\mathbf{A}|\tilde{\mathbf{A}})p(\tilde{\mathbf{A}})\, d\tilde{\mathbf{A}} \qquad p(\tilde{\mathbf{A}}) = \int p(\tilde{\mathbf{A}}|Z)p(Z)\, dZ,$$

where $Z_{1\times d}$ is a graph latent representation [24][Sec. 9.1] with associated prior $p(Z)$. The marginal distribution over graph statistics is:

$$p_\theta(\boldsymbol{\mathcal{I}}) = \int \int p_\theta(\boldsymbol{\mathcal{I}}|\tilde{\mathbf{A}})p_\theta(\tilde{\mathbf{A}}|Z)p(Z)\, d\tilde{\mathbf{A}}\, dZ. \tag{2}$$

We assume graph statistics are independent given the probabilistic matrix $\tilde{\mathbf{A}}$ and model the conditional distribution of each graph statistic as a Gaussian with diagonal variance parameter $\sigma_m^2$,

$$p_\theta(\mathcal{I}_1, \ldots, \mathcal{I}_M|\tilde{\mathbf{A}}) = \prod_{m=1}^{M} \mathcal{N}(\mathcal{I}_m|\phi_m(\tilde{\mathbf{A}}), \sigma_m^2 I). \tag{3}$$

Given the conditional distribution Equation (3), the marginal distribution Equation (2) is a mixture of Gaussians. Figure 2a illustrates the probabilistic graphical model of *GenStat*.

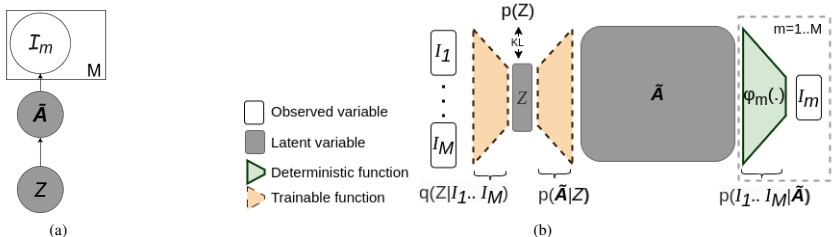

(a)                                                                                    (b)

Figure 2: (a) The proposed latent variable model. The diagram shows the dependency of graph statistics $\{\mathcal{I}_m^i\}_{m=1}^{M}$ on the **latent** probabilistic adjacency $\tilde{\mathbf{A}}$ and graph-level **latent** representation $\mathbf{Z}$. (b) The model's overall architecture.

## 3.2   Training and variational lower bound

For an i.i.d. sample of graphs with associated statistics, $\boldsymbol{\mathcal{I}}^1, \ldots, \boldsymbol{\mathcal{I}}^S$, the marginal log-likelihood is the sum of the marginal log-likelihoods of the individual graphs, $\log p_\theta(\boldsymbol{\mathcal{I}}^1, \ldots, \boldsymbol{\mathcal{I}}^S) = \sum_i \log p_\theta(\boldsymbol{\mathcal{I}}^i)$. The double integral in Equation (2) is generally intractable. We approximate the mixture integral with a Variational Auto-Encoder (VAE) to minimize the negative ELBO as an approximation of the negative log-likelihood for graph statistics:

**Proposition 1.** *Let $p_\theta$ be the marginal likelihood defined in Equation* (2). *Then*

$$-\ln p_\theta(\boldsymbol{\mathcal{I}}^i) = -\ln p_\theta(\mathcal{I}_1^i, \ldots, \mathcal{I}_M^i) \leq E_{Z \sim q_\theta(Z|\mathcal{I}_1^i, \ldots, \mathcal{I}_M^i)} \Big[ -\ln \int p_\theta(\mathcal{I}_1^i, \ldots, \mathcal{I}_M^i|\tilde{\mathbf{A}})p_\theta(\tilde{\mathbf{A}}|Z)d\tilde{\mathbf{A}} \Big]$$
$$+ KL(q_\theta(Z|\mathcal{I}_1^i, \ldots, \mathcal{I}_M^i)||p(Z)). \tag{4}$$

The proof is in the Appendix Section 7.1.1. We approximate the inner integral with a Monte Carlo estimate of the expectation of $p_\theta(\mathcal{I}_1^i, \ldots, \mathcal{I}_M^i|\tilde{\mathbf{A}})$ w.r.t. $p_\theta(\tilde{\mathbf{A}}|Z)$ as follows:

$$\int p_\theta(\mathcal{I}_1^i, \ldots, \mathcal{I}_M^i|\tilde{\mathbf{A}})p_\theta(\tilde{\mathbf{A}}|Z)d\tilde{\mathbf{A}} = E_{p_\theta(\tilde{\mathbf{A}}|Z)}[p_\theta(\mathcal{I}_1^i, \ldots, \mathcal{I}_M^i|\tilde{\mathbf{A}})]$$

$$\approx \frac{1}{T} \sum_{t=1}^{T} p_\theta(\mathcal{I}_1^i, \ldots, \mathcal{I}_M^i|\tilde{\mathbf{A}}^t) \quad where \quad \tilde{\mathbf{A}}^t \sim p_\theta(\tilde{\mathbf{A}}|Z), \tag{5}$$

where the conditional distribution of the underlying probabilistic adjacency matrix can be modelled as a Beta distribution [44].

We implement the *GenStat* objective (4) with an Auto-Encoder in which fully connected neural networks (FCNNs) are used to jointly learn $q_\theta(Z|\mathcal{I}_1^i, \ldots, \mathcal{I}_M^i)$ and $p_\theta(\tilde{\mathbf{A}}|Z)$ with learning parameters $\theta$. Figure 2b illustrates the VAE design of *GenStat*. The prior $p(Z)$ is a standard normal distribution. The variational posterior $q_\theta(Z|\mathcal{I}_1^i, \ldots, \mathcal{I}_M^i)$ is a factored Gaussian with vector mean and variance. See Appendix Section 7.2 for the neural network design and more details concerning implementation.

### 3.3 Permutation-invariant graph generation

Given permutation-invariant statistics, *GenStat* training is permutation-invariant in the sense that the gradient updates are permutation-invariant [76]. Because training is permutation-invariant, the model distribution $p_\theta(\mathbf{A})$ from Equation (1) can be made permutation-invariant by applying a uniform random permutation to the output of the decoder. In addition to generating synthetic graphs, a major use of statistics-based generative models is to define an *inference* distribution to support prediction tasks like link prediction and graph classification [60]. These inference models apply to graph distributions that assign the same probability to graphs with the same statistics, which is appropriate when the graph model is based on statistics only. Accordingly, we define the following permutation-invariant inference distribution for a *GenStat* model:

$$P_\theta(\mathbf{A}) = p_\theta(\mathbf{\Phi}(\mathbf{A}))/C_{\mathbf{\Phi}(\mathbf{A})}, \tag{6}$$

where the distribution $p_\theta(\mathcal{I})$ over statistics follows Equation (2) and $C_{\mathbf{\Phi}(\mathbf{A})}$ is the number of graphs that generate the same statistics as $\mathbf{A}$ (i.e., $C_{\mathbf{\Phi}(\mathbf{A})} = |\{\mathbf{A}' : \mathbf{\Phi}(\mathbf{A}') = \mathbf{\Phi}(\mathbf{A})\}|$).

The following statement summarizes these observations. Appendix Section 7.1.2 provides a proof.

**Observation 1.** *Suppose that a GGM parameterized by $\theta$ is trained with the GenStat architecture and permutation-invariant descriptor functions $\mathbf{\Phi}$. Then the following hold.*

1. *The gradient updates of $\theta$ given a training graph $\mathbf{A}^i$ are permutation-invariant.*

2. *The model distribution $p_\theta(\mathbf{A})$ is permutation-invariant if the generated adjacency matrix is computed by applying a random permutation to the GenStat output.*

3. *The inference distribution $P_\theta(\mathbf{A})$ in Equation (6) is permutation-invariant.*

### 3.4 *GenStat* privacy analysis

This section shows that training the *GenStat* architecture on statistics collected under an LDP guarantee, also satisfies LDP. We employ a graph LDP concept known as Edge LDP [58, 78], which guarantees plausible deniability for the inclusion or removal of a particular edge associated with an individual (node). The neighbor list of $u$ is an $n$-dimensional vector $l_u = [l_{u1}, \ldots, l_{un}]$, with $l_{uv} \in \{0, 1\}$ and $l_{uv} = 1$ iff $u$ and $v$ are connected.

**Definition 1** ($\epsilon$-Edge LDP)**.** *A randomized algorithm $R$ satisfies $\epsilon$-Edge LDP if and only if, for any two neighbour lists $l$ and $l'$, such that $l$ and $l'$ only differ in one bit, and for any output value $s \in range(R)$, the following inequality holds,*

$$p(R(l) = s) \leq e^\epsilon p(R(l') = s). \tag{7}$$

An example of a randomized algorithm $R$ is a perturbed node degree: A user calculates her true node degree, adds noise with a perturbation mechanism (e.g., the Laplace mechanism [13]), and sends the resulting noisy degree to a data curator. The curator aggregates the perturbed node degrees as local graph statistics that can be used to train statistics-based GGMs, such as *GenStat*. The following proposition, with proof in Appendix Section 7.1.3, captures the LDP-preserving properties of *GenStat*.

**Proposition 2.** *Let $\mathbf{R} = (R_1 ... R_M)$ be a set of independent randomized algorithms, outputting perturbed node-level statistics, such that algorithm $R_m$ satisfies $\epsilon_m$-Edge LDP. Then a GenStat GGM trained on the outputs of $\mathbf{R}$ satisfies $\sum_{m=1}^M \epsilon_m$-Edge LDP.*

## 3.5 Observed graph statistics

Graph statistics used in a *GenStat* model are computed by permutation-invariant descriptor functions $\phi(\tilde{\mathbf{A}})$ that can be applied to probabilistic as well as binary adjacency matrices, and are differentiable with respect to entries in the latent adjacency matrix $\tilde{\mathbf{A}}$. To support local privacy, we use graph statistics that aggregate node-level statistics:

$$F(\mathcal{G}) = AGG(\{f(\mathcal{G}_u) : u \in \{1, 2, \ldots, n\}\}), \tag{8}$$

where $\mathcal{G}_u$ denotes the node $k$-hop ego network of node $u$ in graph $\mathcal{G}$ with $n$ nodes, the local statistic $f(\cdot)$ returns a real-valued vector for an ego-graph, and $AGG(\cdot)$ is a permutation invariant function (e.g., average, sum, histogram) that takes as input the set of statistics (e.g., node degree) of the ego-graphs and summarizes them into a vector representation. When $\mathbf{A}$ represents the graph $\mathcal{G}$, we have that $\phi(\mathbf{A}) = F(\mathcal{G})$ is differentiable if the $f(\cdot)$ and $AGG(\cdot)$ functions are differentiable.

The representation of graphs based on local ego-graph properties has been widely studied [22, 64, 79, 83]. Recent studies proposed algorithms for estimating $k$-hop-based graph structural properties under (L)DP [7, 17, 29, 30, 61, 66, 78, 81]. Our experiments use three histogram-based $k$-hop graph statistics: 1) triangle histogram; 2) $k$-HOP neighbors histogram; 3) histogram of random walks, and graph size. We use a differentiable soft histogram [77]; see Section 7.3. The histogram function $h(\cdot)$ transforms a vector of counts into a real-valued soft histogram. We explain each graph statistic next.

*Triangle histogram.* A triangle histogram counts the number of nodes that participate in a given number of triangles. The descriptor function is $h(\frac{1}{2}(\mathbf{A}^3)_{u,u})$.

*$k$-HOP neighbors histogram* for $k = 1, 2, 3, 4$. The $k$-HOP neighbor histogram counts the number of nodes that have a given number of $k$-HOP neighbors. The 1-HOP neighbour histogram is equivalent to the degree histogram. The $k$-Hop descriptor function is $h(\sum(\min(\sum_k \mathbf{A}^k, 1))$ where $\min(\mathbf{A}, 1)_{u,v} = \min(\mathbf{A}_{u,v}, 1)$ and $\sum(\mathbf{A})_u = \sum_v \mathbf{A}_{u,v}$ .

*Random walk histogram of length $r$ for $r = 2, 3$.* The number of walks of length $r$ between node $u$ and $v$ is given by $(\mathbf{A}^r)_{u,v}$ . The random walk histogram of length $r$ counts the number of node pairs with a given number of walks of length $r$ connecting them. It is defined by $h(\mathbf{A}^r)$ [65].

*Graph size.* The size of a graph is its number of edges given by $\frac{1}{2}\sum_v \sum_u \mathbf{A}_{v,u}$.

This paper uses these four statistics in the experiments as *default statistics* and demonstrates that the default statistics are capable of modelling graphs with a wide range of structural characteristics; see Section 4. The default statistics are known from prior research to be generally important for graph modelling across different domains and are easy to interpret [24]. Different graph statistics are important for different applications [55]. The default statistics can be combined with other statistics of interest in a specific application.

## 4 Empirical evaluation

This section compares *GenStat* with parametric and Deep GGMs. Our design closely follows previous studies on generating realistic graphs [6, 88, 89]. We report qualitative and quantitative evaluations of the generated graphs' quality. 1) We compare the performance of *GenStat* with popular statistics-based GGMs. 2) We compare *GenStat* with popular deep GGMs. These models require access to all node interactions, so this is not an apple-to-apples comparison. It measures how much graph generation quality is lost when generation is based on aggregated local statistics, rather than node interactions. 3) We compare GGMs trained under different Edge LDP budgets. 4) We compare GGMs in terms of their benchmark effectiveness for GNNs on link prediction as a downstream task. 5) We also compare deep GGMs and *GenStat* in terms of generation and training time.

**Comparison methods.** We compare *GenStat* to statistics-based and deep adjacency-based GGMs. *Statistics-based baselines.* Statistics-based GGMs that have been used with LDP guarantees [78] include the Chung-Lu Model [5] and the Block Two-Level Erdos-Renyi Model (BTER) [67]. We also evaluate the Stochastic Block Model (SBM) [1] and the Erdos-Renyi model [18], which have been employed in previous comparisons with deep GGMs [88]. See Appendix Section 7.4 for further details.
*Deep adjacency-based baselines* include GraphVAE-MM [89], BiGG [6], GRAN [43] and GraphRNN [88]. To our knowledge, these are the SOTA models for generating realistic graphs.

**Datasets.** Following previous studies, we use real and synthetic datasets [6, 88, 89]. As the datasets are not new, we describe them briefly. We evaluate our model on 3 synthetic datasets: the Lobster trees (Lobster), Grid [88] and Triangle-Grid [89] datasets, all of which consist of graphs with regular structures. We also evaluate our models on 5 real datasets: ogbg-molbbbp (ogbg-mol) [27], Protein [12], IMDb [84], PTC [73] and MUTAG [8]. We randomly partitioned the datasets into train (70%), validation (10%), and test (20%) sets [6, 88, 89]. See Appendix Section 7.5 for detail.

**Evaluating the generated graphs.** Evaluating the sample quality of GGMs requires a comparison between two sets of graphs — the generated graphs and the (held-out) test sets [6, 88, 89]. *Qualitative evaluation* compares the generated graphs by visual inspection.
*Quantitative approaches* compare the distance between the distribution of test graphs and the generated graphs. We use GNN-based [68, 72] and statistics-based [88] metrics to measure the *fidelity (realism)* and the *diversity* of generated graphs. GNN-based metrics extract graph representations with a *reference GNN* that is independent of graph statistics. The reference GNN is either randomly initialized (Random-GNN) [72] or contrastively trained (Pretrained-GNN) [68]. Evaluation metrics then compute the discrepancy between the test set representations and the generated set representations; we report *F1 PR* and *MMD RBF*. Statistic-based evaluation metrics compute the MMD between the test set and generated set with respect to structural properties (orbit counts, degree coefficients, clustering coefficients and diameter) [43]. For all GNN-based metrics, we used 10 GNNs with different random initializations and reported mean $\pm$ standard deviation across different GNNs [72]. Following O'Bray et al. [55] we report scores computed from a 50/50 split of the data sets as the **ideal score**. As in previous work, all models are trained with one random weight initialization to keep the training time feasible [6, 42, 88, 89]. Each trained model is used to generate $S'$ new graphs to compare them with the $S'$ graphs in the test set. See Appendix Section 7.6 for further detail.

## 4.1 Comparison of *GenStat* with statistics-based GGMs on graph realism.

We compare *GenStat* to non-neural baseline models that are widely used in network science. **Qualitative evaluation.** Figure 3 is a visual comparison of randomly selected test and generated graphs graphs for statistics-based GGMs. The *GenStat* graphs are visually much more similar to the test samples than the baseline graphs. For example, instead of Lobster trees, baselines often generate samples that contain multiple cycles. IMDb graphs exhibit a community structure where multiple small communities are interconnected through a central node. For instance, the left IMDb test sample in Figure 3 consists of six communities. None of the baselines was able to generate a graph with high modularity. The Protein samples generated by the baselines exhibit a higher density compared to the test samples. The *GenStat* repository contains the complete collection of generated graphs.

**Quantitative evaluation.** The quantitative evaluation confirms that the quality of graphs generated by *GenStat* is substantially higher. Table 1 reports the *GNN-based MMD RBF* scores. *GenStat* achieved better MMD RBF scores than the baseline GGMs in 12 out of 16 cases. Notably, *GenStat* generated samples have MMD RBF scores up to *ten times* smaller than the baselines for the Random-GNN and Pretrained-GNN approaches. Appendix Table 3 reports the *GNN-based F1 PR*. The F1 PR of generated graphs by *GenStat* is substantially higher than those of the baselines. For example, on Protein, the *GenStat* F1 PR score is 83.72% vs. 71.08% for the next best method with Random-GNN, and 79.95% vs. 23.76% for Pretrained-GNN. Appendix Table 4 shows the results for the *statistics-based* evaluation metrics. *GenStat* generates graphs with up to 1-2 orders of magnitude better statistics-based MMDs, at least in one of the reported MMDs, on almost all datasets.

## 4.2 Comparison of *GenStat* with deep adjacency-based GGMs on graph realism

**Qualitative evaluation.** Appendix Figure 7 provides a visual comparison of the graphs generated by *GenStat* and deep SOTA adjacency-based GGMs. On the real-world datasets, the GRAN graphs are less realistic than the *GenStat* graphs. For the other comparison methods, the quality of their graphs is visually indistinguishable from those generated by *GenStat*.

**Quantitative evaluation.** Table 2 compares *GenStat* with *adjacency-based* GGMs in terms of the GGN-based MMD RBF score of the generated graphs. Although *GenStat* is limited to graph statistics, *for real-world datasets*, it ranks among the top two models in 9 out of 10 reported MMD RBF scores. On the synthetic datasets the *GenStat* scores are competitive with the GraphRNN and GRAN scores but do not reach the level of BiGG and GraphVAE-MM. The difference is that the synthetic datasets

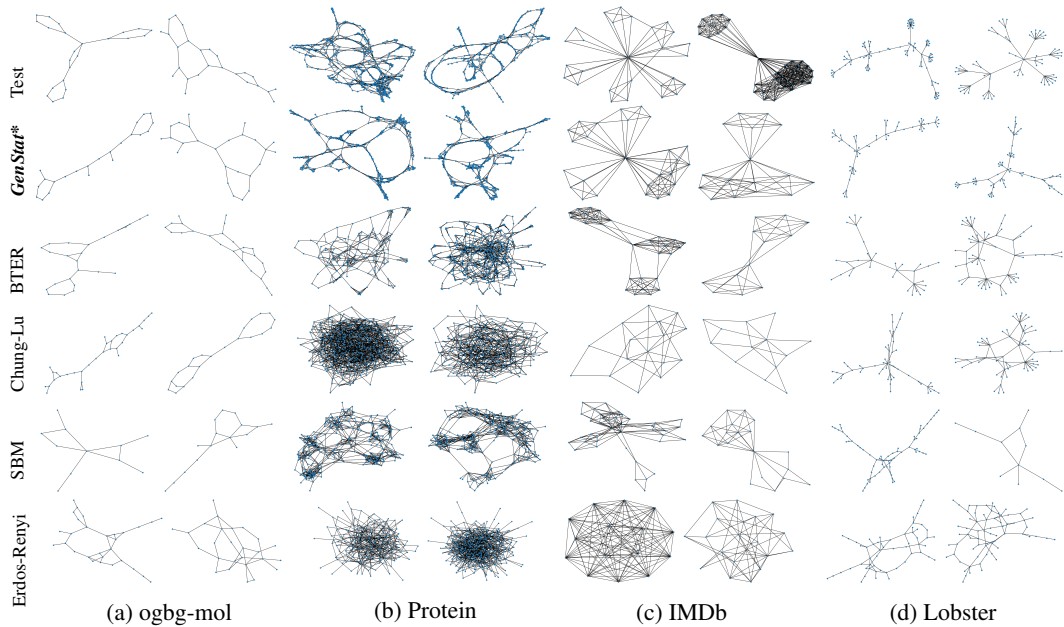

(a) ogbg-mol      (b) Protein      (c) IMDb      (d) Lobster

Figure 3: Visualization of generated graphs by the *statistics-based GGMs*. The top row shows randomly selected graphs from the test set for each dataset, with *varying structural characteristics*. The other rows show graphs generated by each model. The generated graphs shown are the two visually most similar samples in the generated set.

Table 1: Comparison of *GenStat* with *statistics-based* GGMs for the GNN-Based *MMD RBF* score (lower is better). The best result is in bold and the second best is underlined.

| Dataset | Descriptor | 50/50 split | *GenStat** | BTER | Chung-Lu | Erdos-Roni | SBM |
|---|---|---|---|---|---|---|---|
| **ogbg-mol** | Random-GNN | $0.00 \pm 0.00$ | $\mathbf{0.05 \pm 0.01}$ | $0.21 \pm 0.12$ | $\mathbf{0.05 \pm 0.00}$ | $0.12 \pm 0.07$ | $\underline{0.10 \pm 0.03}$ |
| | Pretrained-GNN | $0.00 \pm 0.00$ | $\mathbf{0.01 \pm 0.01}$ | $\underline{0.07 \pm 0.00}$ | $\underline{0.07 \pm 0.00}$ | $0.22 \pm 0.02$ | $0.21 \pm 0.03$ |
| **Protein** | Random-GNN | $0.00 \pm 0.00$ | $\mathbf{0.00 \pm 0.00}$ | $\underline{0.06 \pm 0.02}$ | $0.64 \pm 0.40$ | $0.59 \pm 0.36$ | $0.27 \pm 0.13$ |
| | Pretrained-GNN | $0.01 \pm 0.00$ | $\mathbf{0.00 \pm 0.00}$ | $\underline{0.21 \pm 0.04}$ | $0.60 \pm 0.16$ | $0.59 \pm 0.07$ | $0.50 \pm 0.04$ |
| **IMDb** | Random-GNN | $0.00 \pm 0.00$ | $\mathbf{0.05 \pm 0.03}$ | $0.20 \pm 0.15$ | $0.23 \pm 0.15$ | $0.16 \pm 0.14$ | $0.12 \pm \mathbf{0.03}$ |
| | Pretrained-GNN | $0.01 \pm 0.00$ | $\underline{0.08 \pm 0.02}$ | $\mathbf{0.07 \pm 0.02}$ | $0.27 \pm 0.07$ | $0.16 \pm 0.06$ | $0.20 \pm 0.09$ |
| **PTC** | Random-GNN | $0.01 \pm 0.00$ | $\mathbf{0.00 \pm 0.00}$ | $\underline{0.04 \pm 0.00}$ | $0.09 \pm 0.00$ | $0.10 \pm 0.04$ | $0.19 \pm 0.04$ |
| | Pretrained-GNN | $0.02 \pm 0.00$ | $\mathbf{0.00 \pm 0.00}$ | $\underline{0.08 \pm 0.01}$ | $0.12 \pm 0.01$ | $0.17 \pm 0.02$ | $0.18 \pm 0.02$ |
| **MUTAG** | Random-GNN | $0.00 \pm 0.00$ | $\mathbf{0.05 \pm 0.05}$ | $\underline{0.25 \pm 0.05}$ | $0.27 \pm 0.07$ | $0.37 \pm 0.21$ | $0.27 \pm 0.16$ |
| | Pretrained-GNN | $0.01 \pm 0.00$ | $\mathbf{0.00 \pm 0.00}$ | $\underline{0.18 \pm 0.02}$ | $0.24 \pm 0.03$ | $0.41 \pm 0.03$ | $0.35 \pm 0.06$ |
| **Lobster** | Random-GNN | $0.03 \pm 0.00$ | $0.35 \pm 0.10$ | $\underline{0.15 \pm 0.04}$ | $\mathbf{0.14 \pm 0.02}$ | $0.44 \pm 0.10$ | $0.22 \pm 0.08$ |
| | Pretrained-GNN | $0.10 \pm 0.00$ | $\mathbf{0.04 \pm 0.03}$ | $0.17 \pm 0.01$ | $\underline{0.11 \pm 0.06}$ | $0.32 \pm 0.03$ | $0.27 \pm 0.03$ |
| **Grid** | Random-GNN | $0.02 \pm 0.00$ | $0.53 \pm 0.28$ | $\underline{0.41 \pm 0.28}$ | $0.59 \pm 0.45$ | $\mathbf{0.40 \pm 0.27}$ | $0.61 \pm 0.36$ |
| | Pretrained-GNN | $0.10 \pm 0.00$ | $\mathbf{1.17 \pm 0.10}$ | $\underline{1.21 \pm 0.13}$ | $1.39 \pm 0.09$ | $\underline{1.21 \pm 0.13}$ | $1.25 \pm 0.09$ |
| **Triangle Grid** | Random-GNN | $0.00 \pm 0.00$ | $\mathbf{0.33 \pm 0.07}$ | $\underline{0.38 \pm 0.29}$ | $1.18 \pm 0.29$ | $1.10 \pm 0.27$ | $0.78 \pm 0.34$ |
| | Pretrained-GNN | $0.03 \pm 0.00$ | $\underline{1.11 \pm 0.13}$ | $\mathbf{0.80 \pm 0.17}$ | $1.30 \pm 0.14$ | $1.20 \pm 0.12$ | $1.20 \pm 0.11$ |

exhibit highly regular local structure (e.g., a grid pattern). We observed similar results for the F1 PR and statistics-based scores (Appendix Tables 5 and 6). Apparently the *global* graph statistics used by *GenStat* are insufficient for expressing strict *local* constraints.

In sum, *our deep statistics-based GGM GenStat outperforms previous parametric statistics-based GGMs by orders of magnitude.* In an apples-to-oranges comparison with deep GGMs trained on complete adjacency matrices, it is *competitive or superior on real-world datasets*, but not on synthetic datasets with highly regular local structures such as grid patterns.

## 4.3 Graph realism and edge local differential privacy

We examine training both statistics-based and adjacency-based GGMs with an Edge LDP guarantee. For adjacency-based GGMs, the randomized neighbour list (RNL) approach [58] can be used. Each node randomly flips each bit in its neighbour list with probability $\frac{1}{1+e^\epsilon}$ and sends the perturbed neighbour list to the untrusted curator. A adjacency-based GGM can then be trained on the collected perturbed adjacency matrix. For an $\epsilon$-Edge LDP guarantee, *GenStat* was trained on local ego-graph (node-level) statistics, each perturbed by the Laplace mechanism with variance $\frac{1}{(M\epsilon)^2}$ [14].

Table 2: Comparison of *GenStat* with *adjacency-based* GGMs for the GNN-Based *MMD RBF* score (lower is better). The best result is in bold and the second best is underlined.

| Dataset | Descriptor | 50/50 split | *GenStat* | GraphVAE-MM | BiGG | GRAN | GraphRNN-S | GraphRNN |
|---------|------------|-------------|-----------|-------------|------|------|------------|----------|
| **ogbg-mol** | Random-GNN | $0.00 \pm 0.00$ | $0.05 \pm 0.01$ | **$0.01 \pm 0.00$** | $\underline{0.04 \pm 0.00}$ | $0.44 \pm 0.02$ | $0.44 \pm 0.15$ | $1.53 \pm 0.03$ |
| | Pretrained-GNN | $0.00 \pm 0.00$ | **$0.01 \pm 0.01$** | $0.04 \pm 0.00$ | $\underline{0.03 \pm 0.00}$ | $0.25 \pm 0.03$ | $0.54 \pm 0.04$ | $0.86 \pm 0.09$ |
| **Protein** | Random-GNN | $0.00 \pm 0.00$ | **$0.00 \pm 0.01$** | $0.06 \pm 0.01$ | $0.17 \pm 0.07$ | $\underline{0.05 \pm 0.02}$ | $0.56 \pm 0.18$ | $1.43 \pm 0.32$ |
| | Pretrained-GNN | $0.01 \pm 0.00$ | **$0.00 \pm 0.00$** | $0.17 \pm 0.01$ | $\underline{0.11 \pm 0.00}$ | $0.11 \pm 0.02$ | $0.97 \pm 0.06$ | $1.73 \pm 0.21$ |
| **IMDb** | Random-GNN | $0.00 \pm 0.00$ | $\underline{0.05 \pm 0.03}$ | $0.08 \pm 0.02$ | **$0.02 \pm 0.00$** | **$0.02 \pm 0.00$** | $1.45 \pm 0.03$ | $0.99 \pm 0.06$ |
| | Pretrained-GNN | $0.01 \pm 0.00$ | $\underline{0.08 \pm 0.02}$ | $0.09 \pm 0.03$ | **$0.03 \pm 0.00$** | $0.42 \pm 0.27$ | $1.2 \pm 0.23$ | $0.75 \pm 0.12$ |
| **PTC** | Random-GNN | $0.01 \pm 0.00$ | **$0.00 \pm 0.00$** | $\underline{0.03 \pm 0.00}$ | $0.03 \pm 0.00$ | $0.14 \pm 0.03$ | $0.67 \pm 0.11$ | $0.81 \pm 0.16$ |
| | Pretrained-GNN | $0.02 \pm 0.00$ | **$0.00 \pm 0.00$** | $0.06 \pm 0.00$ | $\underline{0.03 \pm 0.00}$ | $0.18 \pm 0.03$ | $0.53 \pm 0.03$ | $0.51 \pm 0.11$ |
| **MUTAG** | Random-GNN | $0.00 \pm 0.00$ | $0.05 \pm 0.05$ | $0.09 \pm 0.04$ | **$0.03 \pm 0.00$** | $0.09 \pm 0.00$ | $0.53 \pm 0.12$ | $1.08 \pm 0.05$ |
| | Pretrained-GNN | $0.01 \pm 0.00$ | **$0.00 \pm 0.00$** | $0.13 \pm 0.03$ | $\underline{0.09 \pm 0.01}$ | $0.10 \pm 0.01$ | $0.50 \pm 0.14$ | $0.31 \pm 0.05$ |
| **Lobster** | Random-GNN | $0.03 \pm 0.00$ | $0.35 \pm 0.10$ | **$0.09 \pm 0.00$** | $\underline{0.11 \pm 0.00}$ | $0.16 \pm 0.07$ | $0.86 \pm 0.08$ | $0.62 \pm 0.03$ |
| | Pretrained-GNN | $0.10 \pm 0.00$ | **$0.04 \pm 0.03$** | $\underline{0.10 \pm 0.00}$ | $0.11 \pm 0.00$ | $0.29 \pm 0.03$ | $0.74 \pm 0.07$ | $0.22 \pm 0.01$ |
| **Grid** | Random-GNN | $0.02 \pm 0.00$ | $0.53 \pm 0.28$ | **$0.14 \pm 0.01$** | $0.35 \pm 0.00$ | $0.40 \pm 0.00$ | $0.92 \pm 0.05$ | $1.04 \pm 0.09$ |
| | Pretrained-GNN | $0.10 \pm 0.00$ | $1.17 \pm 0.10$ | $\underline{0.35 \pm 0.14}$ | **$0.29 \pm 0.10$** | $0.45 \pm 0.09$ | $1.27 \pm 0.07$ | $1.06 \pm 0.05$ |
| **Triangle Grid** | Random-GNN | $0.00 \pm 0.00$ | $\underline{0.33 \pm 0.07}$ | **$0.18 \pm 0.00$** | $0.38 \pm 0.10$ | $0.31 \pm 0.18$ | $0.79 \pm 0.14$ | $0.94 \pm 0.08$ |
| | Pretrained-GNN | $0.03 \pm 0.00$ | $1.11 \pm 0.13$ | **$0.15 \pm 0.02$** | $\underline{0.34 \pm 0.07}$ | $0.42 \pm 0.27$ | $0.88 \pm 0.16$ | $0.96 \pm 0.10$ |

We compare the quality of graphs generated by *GenStat* and BiGG, the best adjacency-based GGM, against ground-truth test graphs, when each is trained under differential privacy. We also compare perturbed adjacency matrices with the test graphs. This helps us understand the optimal score that can be achieved by an adjacency-based GGM trained on the perturbed adjacency matrices. Graph statistics used in the *GenStat* model for this experiment are triangle histogram and degree histogram. Figure 4 compares the quality scores under the edge $\epsilon$-LDP guarantee, for $\epsilon \in \{0.1, 0.5, 1, 2, 3, 4\}$, using the three datasets on which BiGG showed the biggest advantage over *GenStat* when trained without perturbations (Section 4.2); see the Appendix for other datasets. *GenStat generates more realistic graphs for almost all $\epsilon$-privacy budgets, as indicated by much lower MMD-RBF scores.*

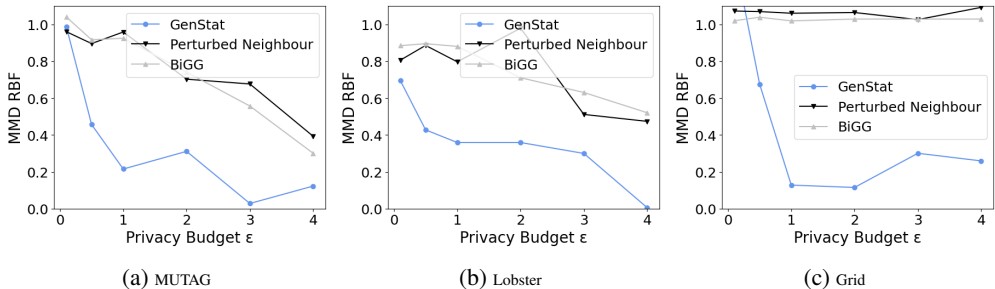

(a) MUTAG       (b) Lobster       (c) Grid

Figure 4: Comparison of *GenStat* (statistics-based GGM) with *BiGG (adjacency-based GGM) under $\epsilon$-Edge LDP guarantee, in terms of the Random GNN-Based MMD RBF score. A lower score is better. The lower bound $\epsilon = 0$ ensures perfect privacy.*

## 4.4 Benchmark effectiveness

A use case for statistics-based GGMs is generating privacy-controlled synthetic graphs for benchmarking the performance of GNNs on downstream tasks, when privacy concerns limit access to the original adjacency matrices in a data collector. The idea behind benchmark effectiveness is that "performance rankings among $m$ GNN models on generated graphs should be similar to the rankings among the same $m$ GNN models on the original graphs" [87]. The methodology is to evaluate each GNN model on the downstream task twice: first on the original dataset, and second on a synthetic dataset generated by the GGM. The benchmark effectiveness of the GGM is then measured by the correlation between the GNNs' task scores from the original and the task scores from the synthetic data. Following [87], we benchmark GNNs on link prediction; see Section 7.8 for further detail. Table 8 compares the benchmark effectiveness of *GenStat* with that of BiGG, the SOTA adjacency-based GGM. On 4 out of 8 datasets, the *GenStat* benchmark effectiveness is competitive with or superior to the benchmark effectiveness of BiGG. For the other datasets, the benchmark effectiveness of BiGG is better, but the *GenStat* graphs still show a substantive correlation (at least 0.5046 for Pearson correlation). For most of the datasets, the MSE of *GenStat* is smaller than that of BiGG. In our opinion, *the benchmark effectiveness of GenStat for link prediction is impressive,*

*especially considering that the model does not observe specific links during training time, unlike BiGG.*

### 4.5 Generation and training time

Evaluating the edge reconstruction probability is expensive and tends to dominate the training time of adjacency-based GGMs. Training statistics-based methods is therefore generally faster than training adjacency-based methods. Specifically, the training time of *GenStat* is up to two orders of magnitude lower than that of the fastest auto-regressive model, and up to 5 times lower than that of GraphVAE-MM. In terms of generation time, both GraphVAE-MM and *GenStat* are much faster than auto-regressive methods, because they generate graphs all-at-once rather than incrementally. Tables 9 and 10 in the Appendix give a detailed comparison of the train and generation time of deep GGMs.

## 5 Discussion and limitations

Graph generative models have a potential attack surface that reveals sensitive information about individuals. Our model contributes to an effective analysis of network structure while maintaining privacy guarantees and minimizing access to the sensitive information of individuals. We expect the *social impact* of our work to be positive. We discuss the limitations of the *GenStat* +VAE design.

*Attributed and heterogeneous graphs.* Following [6, 88, 89], the reported research studies graph generation issues with relatively simple homogeneous graphs. *GenStat* can be extended to attributed/heterogeneous graphs, where nodes/edges possess attributes, including potentially sensitive ones. The graph descriptors can be defined as functions of both feature matrices and edge tensors.

*Computational complexity.* In our implementation, we used matrix multiplication with $O(N^3)$ complexity to exactly compute the descriptors $\Phi(\cdot)$. Approximating graph statistics [16, 32, 59] and exploiting the sparsity of real-word graphs are promising avenues for scaling to large graphs.

*Neural network design.* Following [70, 89], we used FCNN decoders to generate probabilistic adjacency matrices. The all-at-once parallel edge generation of FCNN decoders enables fast training and generation time [24, Ch.9.1.2]. However, they require specifying a maximum number of nodes, and do not scale to large graphs. These limitations in our current system can be addressed with more scalable decoders, e.g. graph transformers [76].

## 6 Conclusion and future work

A statistics-based graph generative model (GGM) is trained on graph statistics that summarize the graph, rather than a complete adjacency matrix. Non-neural parametric models for statistics-based graph generation have been developed in network science for decades. We have described a new *GenStat* framework, which to our knowledge is the first deep GGM architecture based on statistics. Our main motivation for statistics-based graph generation is to avoid requiring sensitive information from individual network participants, especially in the decentralized setting without a single trusted data curator. We show that if local node-level statistics are collected from individuals under edge local differential privacy (LDP), applying *GenStat* to the statistics preserves LDP. In empirical evaluation on eight datasets, graphs generated by *GenStat* were substantially more realistic than those generated by previous statistics-based methods (e.g., by an order of magnitude on the standard MMD RBF quality metric). On real-world datasets, *GenStat* graphs show competitive quality to SOTA GGMs that are based on the entire adjacency matrix, even though *GenStat* sees only summary statistics. We also show that because *GenStat* learns on compressed graph information, training time is much faster than with adjacency-based methods.

A valuable direction for future work is extending statistics-based graph generation to attributed and heterogeneous graphs. There are several directions for scaling *GenStat* architectures to large graphs, such as approximating expected graph statistics, leveraging graph sparsity, and using more scalable decoders. In terms of application tasks, it would be useful to evaluate how well *GenStat* supports learning for downstream tasks (link prediction, node classification) with an LDP guarantee [87].

In sum, statistics-based graph generation is fast and effective in generating high-quality graphs. It offers a strong option for graph learning that respects individuals privacy.

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
