# 7 Appendix

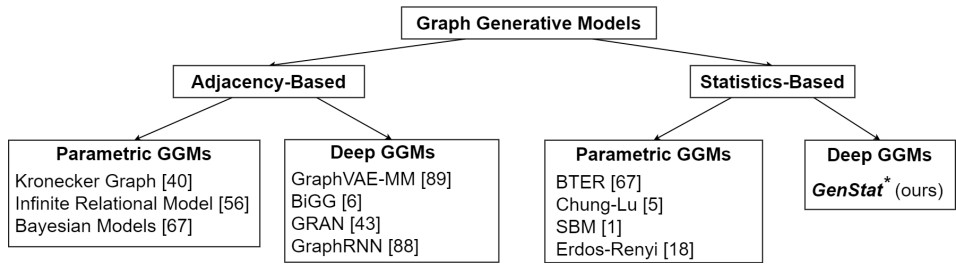

Figure 5: Comparison of *GenStat* architecture to selected graph generative models.

## 7.1 Proofs

### 7.1.1 Proposition 1

Let $p_\theta$ be the marginal likelihood defined in Equation (2). Then

$$-\ln p_\theta(\boldsymbol{\mathcal{I}}^i) = -\ln p_\theta(\mathcal{I}_1^i, \ldots, \mathcal{I}_M^i) \leq E_{Z \sim q_\theta(Z|\mathcal{I}_1^i, \ldots, \mathcal{I}_M^i)} \left[ -\ln \int p_\theta(\mathcal{I}_1^i, \ldots, \mathcal{I}_M^i|\tilde{\mathbf{A}}) p_\theta(\tilde{\mathbf{A}}|Z) d\tilde{\mathbf{A}} \right]$$
$$+ KL(q_\theta(Z|\mathcal{I}_1^i, \ldots, \mathcal{I}_M^i)||p(Z)). \qquad (9)$$

*Proof.*

$$-\ln p_\theta(\boldsymbol{\mathcal{I}}^i) \leq E_{Z \sim q_\theta(Z|\boldsymbol{\mathcal{I}}^i)} \left[ -\ln p_\theta(\boldsymbol{\mathcal{I}}^i|Z) \right] + KL(q_\theta(Z|\boldsymbol{\mathcal{I}}^i)||p(Z))$$
$$= E_{Z \sim q_\theta(Z|\boldsymbol{\mathcal{I}}^i)} \left[ -\ln \int p_\theta(\boldsymbol{\mathcal{I}}^i|\tilde{\mathbf{A}}) p_\theta(\tilde{\mathbf{A}}|Z) d\tilde{\mathbf{A}} \right] + KL(q_\theta(Z|\boldsymbol{\mathcal{I}}^i)||p(Z))$$
$$= E_{Z \sim q_\theta(Z|\mathcal{I}_1^i, \ldots, \mathcal{I}_M^i)} \left[ -\ln \int p_\theta(\mathcal{I}_1^i, \ldots, \mathcal{I}_M^i|\tilde{\mathbf{A}}) p_\theta(\tilde{\mathbf{A}}|Z) d\tilde{\mathbf{A}} \right]$$
$$+ KL(q_\theta(Z|\mathcal{I}_1^i, \ldots, \mathcal{I}_M^i)||p(Z)) \qquad (10)$$

$\square$

### 7.1.2 Observation 1

Suppose that a GGM parameterized by $\theta$ is trained with the *GenStat* architecture and permutation-invariant descriptor functions $\boldsymbol{\Phi}$. Then the following hold.

1. The gradient updates of $\theta$ given a training graph $\mathbf{A}^i$ are permutation-invariant.
2. The model distribution $p_\theta(\mathbf{A})$ is permutation-invariant if the generated adjacency matrix is computed by applying a random permutation to the *GenStat* output.
3. The inference distribution $P_\theta(\mathbf{A})$ in Equation (6) is permutation-invariant.

*Proof.* For training invariance, note that a *GenStat* loss function must depend on the graph statistics only, that is $L_\theta(\mathbf{A}^i) = L_\theta(\phi(\mathbf{A}^i))$ (see Equation (4) for an example). Therefore

$$L_\theta(\mathbf{A}^i) = L_\theta(\phi(\mathbf{A}^i)) = L_\theta(\phi(\mathbf{A}_\pi^i)) = L_\theta(\mathbf{A}_\pi^i).$$

Since the loss function is permutation-invariant, so are its gradient updates, which establishes claim 1.

For the second claim, recall that $p_\theta(\mathbf{A})$ is the generative distribution defined by Equation (1). Adding a random permutation means that the final probability $p^*(\mathbf{A})$ of generating an adjacency matrix $\mathbf{A}$ is essentially the probability of generating a permutation of $\mathbf{A}$:

$$p^*(\mathbf{A}) = \sum_\pi p_\theta(\mathbf{A}_\pi)/n! \qquad (11)$$

The probability $p^*$ is permutation-invariant: if $\mathbf{A}$ is a permutation of $\mathbf{A}_\pi$, then they can be generated by the same adjacency matrices $\mathbf{A}'$ with the same probability.

Let $\mathbf{A}_\pi$ be a permutation of matrix $\mathbf{A}$. Since the statistics are permutation-invariant, we have that $\mathbf{\Phi}(\mathbf{A}) = \mathbf{\Phi}(\mathbf{A}_\pi)$. Therefore

$$P_\theta(\mathbf{A}) = p_\theta(\mathbf{\Phi}(\mathbf{A}))/C_{\mathbf{\Phi}(\mathbf{A})} = p_\theta(\mathbf{\Phi}(\mathbf{A}_\pi))/C_{\mathbf{\Phi}(\mathbf{A}_\pi)} = P_\theta(\mathbf{A}_\pi)$$

which establishes claim 3. $\qquad\square$

### 7.1.3 Proposition 2

Let $\mathbf{R} = (R_1...R_M)$ be a set of independent randomized algorithms, outputting perturbed node-level statistics, such that algorithm $R_m$ satisfies $\epsilon_m$-Edge LDP. Then a *GenStat* GGM trained on the outputs of $\mathbf{R}$ satisfies $\sum_{m=1}^{M} \epsilon_m$-Edge LDP.

*Proof.* This proof uses two properties of LDP: composability and immunity to post-processing [2].

**Lemma 1.** *Collected data (local node-level statistics calculated and perturbed by $\mathbf{R}$) from each node satisfies $\sum_{m=1}^{M} \epsilon_m$-Edge LDP.*

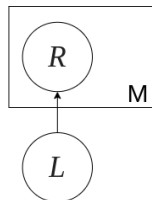

Figure 6: PGM of the randomized algorithms.

*Proof.* Figure 6 illustrates the PGM of Randomized algorithms. Randomized algorithms are independent given the neighbour list $l$,

$$p(R_m(l) = s_m, \; R_{m'}(l) = s_{m'}) = p(R_m(l) = s_m)p(\, R_{m'}(l) = s_{m'}),$$

and as a result,

$$p(\mathbf{R}(l) = \mathbf{s}) = \prod_{m=1}^{M} p(R_m(l) = s_m). \tag{12}$$

Applying the definition of edge differential privacy for each $R_m$,

$$\prod_{m=1}^{M} p(R_m(l) = s_m) \le \prod_{m=1}^{M} e^{\epsilon_m} p(R_m(l') = s_m) = e^{\sum_{m=i}^{M} \epsilon_m} p(\mathbf{R}(l') = \mathbf{s}) \tag{13}$$

where the last equation is the definition of $\sum_{m=1}^{M} \epsilon_m$-Edge LDP. $\qquad\square$

The GGM parameters are a function of the perturbed graph statistics as learning input. The generated graphs are a function of the GGM parameters. Since the generated graphs cannot contain more information than the perturbed graph statistics, immunity to post-processing ($f(R(l))$ guarantees LDP for any function $f$) [2]. $\qquad\square$

### 7.2 Implementation and neural network design detail

The *encoder* consists of 2 layers of FCNNs which follow a Batch Norm layer [31]. The layers are followed by a LeakyReLU activation function and a Layer Normalization [3]. The latent vector $Z$ is normally distributed and its mean and variance are generated from two separate one layer FCNNs over the output of the shared FCNNs.

For further efficiency, we replace the Monte Carlo *decoder* of Equation (5) with a deterministic decoder $\tilde{\mathbf{A}}_Z$. The deterministic decoder is implemented by a trainable FCNN that takes as input a latent graph level representation $Z$. Similar to [4] and [70], the decoder assumes a fixed maximum number of nodes, $N$, where isolated nodes are added to smaller graphs and is parameterized by 4 FCNN, which returns the flattened probabilistic adjacency matrix $\tilde{\mathbf{A}}_Z$ [24, Sec.9.1.2]. Each of the first 3 FCLs are followed by a LeakyReLU activation function and a Layer Normalization [3]. The last layer of the decoder is followed by a sigmoid activation function and the descriptor functions $\mathbf{\Phi} = \{\phi_m()\}_{m=1}^M$. Figure 2b shows the overall architecture.

To estimate the variance $\sigma_m^2$ for a reconstructed statistic $\mathcal{I}_M^i$ (Equation (2)), we use the optimal $\sigma$-VAE method from the calibrated Gaussian framework [62, 89].

In this study, following [43, 88], we focus on undirected unweighted graphs. The implementation can be easily extended to directed graphs. We also use the $\beta$-VAE [26] setting to balance the reconstruction log-likelihood with the KLD term, where $\beta \in \{1, 4, 20\}$. Inspired by Vaswani et al. [74] the model uses Adam optimizer with 1 cycle learning rate schedule where the learning rate varies from $lr/25$ to $lr \in \{0.001, 0.0003, 0.0001\}$, then down to $lr/100$. Hyperparameters are selected by validation set performance. The model is trained for $40,000$ epochs for all the datasets with batch size 64, except for the Protein dataset which is trained for $10,000$ epochs and batch size 32.

*k-HOP neighbors histogram.* The leaky version of the min function handles sparse gradients for longer steps and dense graphs. The leaky min function is defined as follows,

$$\text{LeakyMin}(\mathbf{A}_{uv}^k, 1) = \begin{cases} \mathbf{A}_{uv}^k, & \text{if } \mathbf{A}_{uv}^k \leq 1 \\ \alpha \mathbf{A}_{uv}^k + 1, & \text{otherwise,} \end{cases} \tag{14}$$

where, in our implementation, $\alpha = 0.01$.

**Synthetic Graph Generation.** At test time, following [70], graphs are generated as follows: 1) graph latent representation is sampled from the prior $p(Z)$. 2) The decoder computes a probabilistic latent adjacency matrix $\tilde{\mathbf{A}}$. 3) A 0.5 threshold is applied to convert link probabilities into hard binary links.

### 7.3 Histogram function

In this study, histogram functions are utilized to summarize node-level statistics into a vector representation, ensuring permutation invariance and providing graph-level statistics [9, 53, 57]. This study adopts a differentiable soft histogram function [77] with Gaussian membership function, defined over $B = O(\sqrt{N})$ centers. Following [49], this study adopts a multiple binning model and utilizes equal-width and equal-frequency binning schemas, Algorithm 1 and 2. The bins' centers are calculated in a prepossessing step. The differentiable histogram function is based on a soft assignment of points to bins given the bin centers. In detail, the membership of datapoint $x_i$ to the bin center $C_b$ is given by,

$$\mathcal{M}(x_i) = e^{-\left(\frac{x_i - C_b}{\gamma}\right)^2}, \tag{15}$$

where $\gamma$ is a constant.

---

**Algorithm 1** Equal-Width Binning

---

**Require:** A set of data points $D = x_1, x_2, ..., x_n$, the desired number of bins $B$
**Ensure:** A set of $B$ bins' center $C_1, C_2, ..., C_B$, where each bin contains approximately the same width
  1: $w = (\max(D) - \min(D))/B$                     ▷ Compute the bins' width
  2: **for** $b = 0$ to $B$ **do**
  3:     $C_b = \min(D) + b \cdot w$                 ▷ Define the center of the bin
  4: **end for**
  5: **return** the set of $B$ bins' center $C_1, C_2, ..., C_B$.

---

---
**Algorithm 2** Equal-Frequency Binning
---
**Require:** A set of data points $D = x_1, x_2, ..., x_n$, the desired number of bins $B$
**Ensure:** A set of $B$ bins' center $C_1, C_2, ..., C_B$, where each bin contains approximately the same number of data points
  1: $k = n/B$                                 ▷ Compute the number of data point in each bin
  2: Sort the data points in increasing order: $D' = x'_1, x'_2, ..., x'_n$ where $x'_1 \leq x'_2 \leq ... \leq x'_n$
  3: Assign the first $k$ data points to the first bin $B_1$, the next $k$ data points to the second bin $B_2$, and so on, until all data points have been assigned to a bin.
  4: **for** $b = 0$ to $B$ **do**
  5:      $C_b = mean(B_b)$                             ▷ Define the center of the bin
  6: **end for**
  7: **return** the set of $B$ bins' center $C_1, C_2, ..., C_B$.
---

## 7.4 Baselines

This section briefly describes the graph generative models which we use for benchmarking.

**BTER.** A statistics-based GGM that takes degree and clustering coefficient sequences as sufficient statistics [67].
**Chung-Lu.** A statistics-based GGM that takes the degree sequence as sufficient statistics [5].
**Erdos-Roni.** A statistics-based model that only requires graph density (probability of an edge existing between any pair of nodes) [18].
**SBM.** A statistics-based GGM that requires clustering membership for each node and an edge propensities between clusters [1].
**GraphVAE-MM.** An all-at-once deep GGM that generate the graph in $O(1)$ steps [89].
**BiGG.** Auto-regressive deep GGM that leverages graphs sparsity and generates the graph in $O(\log n)$ steps [6].
**GRAN.** Auto-regressive deep GGM that generates a graph in $O(n)$ steps, a block of nodes and associated edges at step [43].
**GraphRNN.** Auto-regressive deep GGM that generates the graph in $O(n^2)$ steps. Each step generates one entry in the GraphRNN design ($O(n^2)$ steps), or one column in the GraphRNN-S design ($O(n)$ steps) [88].

For statistics-based GGMs we used the public repository provided by [69]. For the deep GGMs we used the original papers' public repository; hence no consent was needed to curate this study. For all baselines, we used the hyper-parameters setting provided by the original papers and trained the models for a maximum of 24 hours.

## 7.5 Datasets

Following previous studies [43, 88, 89], we use synthetic and real graph datasets as follows.

**ogbg-molbbbp (ogbg-mol).** Includes 2039 real-world molecular graphs with $2 \leq |V| \leq 132$ [27].
**Protein.** Includes 918 real-world Protein graphs with $100 \leq |V| \leq 500$ [12].
**IMDb-BINARY (IMDb).** Comprises the ego-graphs of 1000 actors/actresses who played roles in movies in IMDb [84].
**PTC.** Is a dataset of 344 chemical compounds that reports the carcinogenicity of male and female rats [73].
**MUTAG.** MUTAG is a dataset of 188 mutagenic aromatic and heteroaromatic nitro compounds [8].
**Lobster Tree (Lobster).** Consists of 100 synthetic graphs with $10 \leq |V| \leq 100$. Generated using the code from [88].
**Grid.** Consists of 100 synthetic 2D graphs, regular tiling of the 2D plane with equilateral squares, with $100 \leq |V| < 400$ [88].
**Triangle Grid.** Consists of 100 synthetic 2D graphs, regular tiling of the 2D plane with equilateral triangles, with $100 \leq |V| < 400$ [89].

The datasets utilized in this research study do not include any personally identifiable information or offensive/harmful content regarding individuals or communities. These datasets are openly accessible and publicly available.

### 7.6 Evaluation metrics and experimental setting details

**The GNN-based evaluation metrics.** These metrics are based on the representations obtained by GNNs. Thompson et al. [72] suggested utilizing randomly initialized GNNs (Random-GNN), inspired by the ability of GNNs to extract meaningful graph representations without any training. Shirzad et al. [68] proposed using representations from contrastively trained GNNs (Pretrained-GNN), rather than random GNNs, to fully leverage the expressive power of GNNs. Following Shirzad et al. [68] we add higher-order local information of nodes including the degree of a node, three-node and four-node clustering features for each node as explicit features.

**MMD RBF and F1 PR.** The *MMD RBF* metric calculates the MMD of graph representations in two sets and primarily measures the fidelity or realism of generated graphs [88]. *F1 PR* evaluates the percentage of generated graph representations that fall within the manifold of test graph representations (precision) and the percentage of test graph representations that fall within the manifold of generated graph representations (recall). This metric considers the diversity of the generated graphs as a factor [72].

The GNN-based evaluation metrics are computed using the packages specified in the original paper [68, 72]. Statistic-based evaluation metrics are computed using an implementation provided in Liao et al. [43]. Following O'Bray et al. [55] we report scores computed from a 50/50 split of the data sets as an *Ideal score*. For Pretrained-GNN based metrics we trained the GNNs on 60% of the data and reported scores computed from the 50/50 split of the rest of the samples as Ideal score.

### 7.7 Evaluation of GGMs in detail

Table 3 reports the detailed scores for F1 PR, as we reported in the main text, we observe a substantial improvement in comparison with statistic-based GGMs. The F1 PR metrics collapse on synthetic datasets with highly regular structures for Pretrained-GNN. For datasets with highly regular structures, Pretrained-GNNs learn instantly to discriminate the real graphs from the generated ones very easily, and for generated graphs with even very small perturbations measure a substantial distance [68]. The problem was noted in the original paper.

Figure 7 shows a qualitative comparison of *GenStat* with SOTA adjacency-based GGMs on real-world datasets. We do not show results for GraphRNN because it is both qualitatively and quantitatively clearly worse than later methods. The GRAN graphs are the least realistic, and tend to merge different communities, therefore exhibiting fewer communities in the test graph. This is especially visible in the IMDb and Lobster graphs. For synthetic grid datasets (Triangle and Grid) we find that the graphs generated by *GenStat* are visually worse than those generated by adjacency-based models, including GRAN. The grid dataset outputs are not shown in the figure, but are available in the *GenStat*'s repository.

### 7.8 Benchmark effectiveness in detail

In this paper, following Yoon et al. [87] we use link prediction as a downstream task to evaluate GGMs' effectiveness for benchmarking graph neural networks.

*The link prediction task* is to estimate the probability that a pair of nodes in a graph are connected by a link. Following [37, 47], we remove a fraction of edges with the same number of non-existent edges, as a test set, from the input graphs. We test the GNN model predictions of the test set after training on the remaining edges. The test set comprises 20% held-out edges, together with the same number of non-existent edges. For a given dataset, the link prediction experiment is performed 10 times, each using a random train/test split, and the mean of the performance metric is reported.

*The GNN effectiveness* is computed as follows.

1. Train a GGM (e.g., *GenStat*) on the original dataset (e.g., IMDb). Generate a synthetic graph dataset of the same size as the original one (e.g. IMDb-Synth).

2. Score each GNN model (e.g., GCN) on the original graph dataset for the downstream task. E.g., for link prediction, divide links into training and test edges, train the GNN, report test

performance metric (Accuracy is the GNN test metric in Yoon et al. [87]). Call this score *GNN-original*.

3. Score each GNN model (e.g., GCN) on the synthetic graph dataset for the downstream task. Call this score *GNN-synthetic*.

4. For each dataset, report the correlation between the GNN-original and the GNN-synthetic scores. Yoon et al. [87] use Pearson correlation, Spearman rank correlation, and MSE (squared difference beween the accuracy scores).

Table 7 illustrates the idea by showing the link prediction accuracies and the resulting correlations/MSE benchmark effectiveness scores for the Protein dataset. Following [87] we choose popular GNN models for benchmarking: GCN [38], GIN [83], SGC [82], and GAT [75]. Link prediction was performed with a dot product decoder as in [87], and the models were trained for 200 epochs. Table 8 compares the benchmark effectiveness of *GenStat* with that of BiGG, the SOTA adjacency-based GGM. Again this is an apples-to-oranges comparison because BiGG has access to the entire adjacency matrix. In these experiments, we use random random features concatenated with higher-order local information of nodes as the nodes' feature matrices.

## 7.9 Code overview

The *GenStat* implementation is provided at `https://github.com/kiarashza/GenStat.git`. The file "GlobalPrespective.py" includes the training pipeline as well as the implementation of the objective function. Source codes for loading the real and synthetic graphs are included in "data.py". "data/LDP/" contains the perturbed adjacency matrices used for graph generation with LDP guarantee. All the Python packages used in our experiments are provided in "requirement.yml". Generated graph samples for *GenStat* are provided in the "ReportedResult/" directory, both in the pickle and png format. `https://drive.google.com/drive/folders/1mF-kU021-ceNh6ejLgf41sSE9FEzc01Q` contains the generated samples by the baselines. "GNN.py" contains the implementation of the GNN models we used for benchmarking in Section 4.4. We used "GraphGenerationWithLDP.py", "Graph-Generation.py", "BechmarkingGNNs.py" for the *GenStat*'s reported result in the Section 4. These files contain the necessary commands and hyperparameters used in the experiments.

## 7.10 System architecture

The code for all models is run on the same system, an Intel(R) Core(TM) i9-9820X CPU 3.30GHz and Nvidia TITAN RTX GPU with TU102-core. Because of package compatibility issues, GraphRNN(-S) is run on an Intel(R) Core(TM) i7-5820K CPU 3.30GHz and a GM200 GeForce GTX TITAN X.

Table 3: Comparison of *GenStat* with *statistics-based* GGMs for the *GNN-based F1 PR* score (higher is better). The best result is in bold and the second best is underlined. The F1 PR of generated graphs by *GenStat* is substantially higher than those of the baselines. For example, on Protein, the *GenStat* F1 PR score is 92.95% vs. 53.93% for the next best method with Random-GNN, and 87.37% vs. 23.76% for Pretrained-GNN.

| Dataset | Metric | 50/50 split | *GenStat** | BTER | Chung-Lu | Erdos-Renyi | SBM |
|---|---|---|---|---|---|---|---|
| **ogbg-mol** | Random-GNN | 97.63 ± 0.39 | 89.96 ± 4.78 | 65.64 ± 19.45 | **92.67 ± 1.04** | 62.56 ± 10.67 | 61.25 ± 8.24 |
| | Pretrained-GNN | 95.09 ± 0.61 | 68.77 ± 4.43 | 80.89 ± 2.44 | **80.71 ± 3.3** | 43.53 ± 5.59 | 38.16 ± 2.35 |
| **Protein** | Random-GNN | 97.39 ± 0.46 | **83.72 ± 7.05** | 71.08 ± 10.48 | 8.10 ± 14.86 | 2.71 ± 5.58 | 15.46 ± 13.50 |
| | Pretrained-GNN | 95.02 ± 1.14 | **79.95 ± 17.14** | 23.76 ± 8.97 | 0.00 ± 0.00 | 0.00 ± 0.00 | 1.70 ± 2.62 |
| **IMDb** | Random-GNN | 98.75 ± 0.29 | **85.40 ± 3.32** | 53.93 ± 12.35 | 9.35 ± 4.05 | 37.68 ± 10.13 | 48.33 ± 12.50 |
| | Pretrained-GNN | 97.12 ± 0.72 | 71.89 ± 4.50 | **88.47 ± 4.50** | 13.66 ± 8.66 | 41.49 ± 11.88 | 18.78 ± 5.89 |
| **PTC** | Random-GNN | 98.22 ± 0.80 | 94.79 ± 1.78 | 89.14 ± 4.2 | **96.47 ± 2.51** | 68.56 ± 7.95 | 69.93 ± 8.41 |
| | Pretrained-GNN | 98.08 ± 1.67 | **82.35 ± 11.33** | 73.48 ± 4.29 | 76.43 ± 3.03 | 39.98 ± 5.05 | 43.18 ± 4.98 |
| **MUTAG** | Random-GNN | 98.69 ± 0.24 | **82.78 ± 12.31** | 65.64 ± 19.45 | 73.78 ± 14.45 | 30.60 ± 17.31 | 52.11 ± 18.65 |
| | Pretrained-GNN | 96.67 ± 0.7 | 51.98 ± 10.87 | **58.73 ± 8.95** | 53.74 ± 8.32 | 16.9 ± 12.41 | 28.39 ± 6.68 |
| **Lobster** | Random-GNN | 95.48 ± 0.92 | **97.43 ± 0.00** | 82.22 ± 14.13 | 92.35 ± 4.22 | 37.23 ± 16.85 | 64.70 ± 22.43 |
| | Pretrained-GNN | 95.38 ± 2.53 | **89.94 ± 3.68** | 69.78 ± 4.77 | 73.47 ± 8.66 | 43.86 ± 4.48 | 43.59 ± 4.92 |
| **Grid** | Random-GNN | 100.0 ± 0.00 | 50.40 ± 28.42 | **64.00 ± 42.76** | 56.48 ± 40.43 | 56.31 ± 39.43 | 30.44 ± 38.00 |
| | Pretrained-GNN | 100.0 ± 0.00 | 0.00 ± 0.00 | 0.00 ± 0.00 | 0.00 ± 0.00 | 0.00 ± 0.00 | 0.0 ± 0.00 |
| **Triangle Grid** | Random-GNN | 97.8 ± 0.91 | **75.86 ± 22.18** | 59.37 ± 37.23 | 1.33 ± 4.00 | 2.00 ± 4.26 | 18.66 ± 26.26 |
| | Pretrained-GNN | 94.09 ± 2.14 | 0.00 ± 0.00 | 0.00 ± 0.00 | 0.00 ± 0.00 | 0.00 ± 0.00 | 0.00 ± 0.00 |

Table 4: Comparison of *GenStat* with *statistic-based* GGMs for the *statistic-based MMD* score (lower is better). The best result is in bold and the second best is underlined. *GenStat* generates graphs with up to 1-2 orders of magnitude better statistics-based MMDs on almost all datasets.

| Dataset | Metric | 50/50 split | *GenStat\** | BTER | Chung-Lu | Erdos-Renyi | SBM |
|---|---|---|---|---|---|---|---|
| **ogbg-molbbbp** | Deg. | $2e^{-4}$ | 0.004 | **$1e^{-4}$** | 0.001 | 0.029 | 0.028 |
| | Clus. | $2e^{-5}$ | **$3e^{-4}$** | 0.089 | 0.079 | 0.461 | 0.299 |
| | **Orbit**. | $9e^{-5}$ | 0.009 | **0.001** | **0.001** | 0.037 | 0.018 |
| | Spect. | $5e^{-4}$ | 0.015 | **0.013** | **0.013** | 0.022 | 0.023 |
| | Diam. | 0.002 | 0.036 | 0.031 | **0.024** | 0.350 | 0.292 |
| **Protein** | Deg. | $4^{-5}$ | 0.002 | 0.001 | **4e$^{-5}$** | 0.019 | 0.027 |
| | Clu. | 0.004 | **0.072** | 0.097 | 0.606 | 0.519 | 0.242 |
| | Orbit. | $5e^{-4}$ | **0112** | 0.204 | 1.121 | 1.065 | 0.917 |
| | Spec | $4e^{-4}$ | **0.008** | 0.031 | 0.067 | 0.064 | 0.051 |
| | Diam. | 0.003 | **0.012** | 0.78 | 1.151 | 1.108 | 0.511 |
| **IMDb** | Deg. | 0.001 | 0.048 | **0.007** | 0.038 | 0.036 | 0.068 |
| | Clus. | 0.004 | 0.291 | **0.138** | 0.372 | 0.261 | 0.350 |
| | Orbit. | 0.004 | **0.020** | 0.314 | 0.330 | 0.196 | 0.296 |
| | Spect. | 0 | 0.054 | **0.014** | 0.107 | 0.102 | 0.100 |
| | Diam. | $4e^{-5}$ | **0.059** | 0.231 | 0.223 | 0.109 | 0.439 |
| **MUTAG** | Deg. | $3e^{-4}$ | 0.001 | **7e$^{-4}$** | 0.003 | 0.036 | 0.056 |
| | Clus. | 0 | **0.004** | 0.132 | 0.245 | 0.606 | 0.577 |
| | Orbit. | $1e^{-5}$ | **0.002** | **0.002** | 0.005 | 0.046 | 0.014 |
| | Spec. | 0.005 | 0.036 | **0.024** | 0.028 | 0.038 | 0.033 |
| | Diam. | 0.013 | 0.072 | **0.044** | 0.080 | 0.178 | 0.361 |
| **PTC** | Deg. | $1e^{-4}$ | 0.037 | textbf0.009 | 0.015 | 0.113 | 0.107 |
| | Clus. | $9e^{-5}$ | **0.008** | 0.114 | 0.094 | 0.309 | 0.240 |
| | Orbit. | $8e^{-5}$ | 0.005 | **9e$^{-4}$** | 0.002 | 0.011 | 0.002 |
| | Spec. | 0.002 | 0.032 | **0.024** | 0.031 | 0.037 | 0.041 |
| | Diam. | 0.013 | 0.0265 | 0.039 | 0.089 | **0.029** | 0.083 |
| **Triangle Grid** | Deg. | $3e^{-5}$ | 0.075 | **0.002** | 0.007 | 0.223 | 0.239 |
| | Clus. | 0.002 | 1.175 | **0.933** | 1.434 | 1.174 | 1.087 |
| | Orbit. | $8e^{-5}$ | 0.239 | **0.479** | 1.857 | 1.701 | 1.492 |
| | Spec. | 0.004 | **0.036** | 0.058 | 0.094 | 0.091 | 0.083 |
| | Diam. | 0.014 | **0.499** | 1.074 | 1.474 | 1.466 | 1.407 |
| **Grid** | Deg. | $1e^{-5}$ | 0.044 | **5e$^{-4}$** | 0.001 | 0.286 | 0.293 |
| | Clus. | 0 | **0** | 0.147 | 0.245 | 0.754 | **0.033** |
| | Orbit. | $2e^{-5}$ | **0.015** | 0.061 | 0.051 | 0.853 | 0.802 |
| | Spec. | 0.004 | **0.040** | 0.057 | 0.050 | 0.045 | 0.041 |
| | Diam. | 0.014 | **0.314** | 1.288 | 1.447 | 1.097 | 1.113 |
| **Lobster** | Deg. | 0.002 | 0.014 | **0.003** | 0.009 | 0.199 | 0.15 |
| | Clus. | 0 | **0.005** | 0.297 | 0.362 | 0.229 | 0.082 |
| | Orbit. | 0.002 | **0.084** | 0.042 | 0.040 | 0.065 | 0.124 |
| | Spec. | 0.005 | 0.051 | **0.036** | 0.040 | 0.193 | 0.105 |
| | Diam. | 0.032 | 0.335 | 0.181 | 0.082 | 0.080 | **0.069** |

Table 5: Comparison of *GenStat* with *adjacency-based* GGMs for the *GNN-based F1 PR* score (higher is better). Although *GenStat* is limited to graph statistics (less information), *for real-world datasets*, it ranks among the top two models in 8 out of 10 reported F1 PR scores.

| Dataset | Metric | 50/50 split | *GenStat\** | GraphVAE-MM | BiGG | GRAN | GraphRNN-S | GraphRNN |
|---|---|---|---|---|---|---|---|---|
| **ogbg-mol** | Random-GNN | $97.63 \pm 0.39$ | $89.96 \pm 4.78$ | $92.95 \pm 2.05$ | **$95.26 \pm 1.09$** | $89.90 \pm 2.34$ | $49.62 \pm 9.32$ | $16.05 \pm 0.51$ |
| | Pretrained-GNN | $95.09 \pm 0.61$ | $68.77 \pm 4.43$ | $83.57 \pm 1.86$ | **$86.59 \pm 1.20$** | $75.91 \pm 1.68$ | $30.91 \pm 6.33$ | $10.96 \pm 3.32$ |
| **Protein** | Random-GNN | $97.39 \pm 0.46$ | $83.72 \pm 7.05$ | $78.99 \pm 5.75$ | $76.60 \pm 13.4$ | **$93.93 \pm 1.67$** | $39.47 \pm 25.45$ | $0.00 \pm 0.00$ |
| | Pretrained-GNN | $95.02 \pm 1.14$ | **$79.95 \pm 17.14$** | $74.91 \pm 3.6$ | $75.52 \pm 2.03$ | $71.94 \pm 6.32$ | $2.67 \pm 2.80$ | $0.00 \pm 0.0$ |
| **IMDb** | Random-GNN | $98.75 \pm 0.29$ | $85.40 \pm 3.32$ | **$92.39 \pm 3.74$** | $88.74 \pm 1.33$ | $88.92 \pm 1.08$ | $0.00 \pm 0.00$ | $0.95 \pm 2.85$ |
| | Pretrained-GNN | $97.12 \pm 0.72$ | $71.89 \pm 5.72$ | $88.08 \pm 5.16$ | **$90.06 \pm 1.55$** | $88.38 \pm 15.16$ | $0.00 \pm 0.00$ | $0.00 \pm 0.00$ |
| **PTC** | Random-GNN | $98.22 \pm 0.80$ | $94.79 \pm 1.78$ | $87.13 \pm 2.5$ | **$97.28 \pm 0.93$** | $68.38 \pm 8.52$ | $65.63 \pm 13.85$ | $85.10 \pm 6.34$ |
| | Pretrained-GNN | $98.08 \pm 1.67$ | $82.35 \pm 11.33$ | $81.06 \pm 5.47$ | **$92.42 \pm 1.39$** | $42.32 \pm 6.51$ | $68.73 \pm 9.66$ | $70.84 \pm 15.09$ |
| **MUTAG** | Random-GNN | $98.69 \pm 0.24$ | $82.78 \pm 12.31$ | $80.67 \pm 11.73$ | $73.70 \pm 4.52$ | **$84.83 \pm 3.26$** | $17.67 \pm 12.26$ | $0.00 \pm 0.00$ |
| | Pretrained-GNN | $96.67 \pm 0.70$ | $51.98 \pm 10.87$ | $57.68 \pm 7.65$ | **$73.14 \pm 3.22$** | $70.50 \pm 1.87$ | $6.05 \pm 3.63$ | $33.64 \pm 12.04$ |
| **Lobster** | Random-GNN | $95.48 \pm 0.92$ | **$97.43 \pm 0.00$** | $95.36 \pm 2.21$ | $95.77 \pm 2.54$ | $74.12 \pm 12.69$ | $31.18 \pm 28.03$ | $22.00 \pm 6.12$ |
| | Pretrained-GNN | $95.38 \pm 2.53$ | $89.94 \pm 3.68$ | $87.83 \pm 5.72$ | **$97.42 \pm 2.00$** | $48.58 \pm 6.11$ | $10.2 \pm 6.33$ | $83.17 \pm 4.01$ |
| **Grid** | Random-GNN | $100.00 \pm 0.00$ | $50.40 \pm 28.42$ | **$97.46 \pm 5.12$** | $92.16 \pm 0.81$ | $80.27 \pm 2.17$ | $55.63 \pm 28.51$ | $3.15 \pm 4.03$ |
| | Pretrained-GNN | $100.0 \pm 0.00$ | $0.00 \pm 0.00$ | $39.03 \pm 7.02$ | **$74.57 \pm 13.59$** | $41.91 \pm 23.33$ | $0.00 \pm 0.00$ | $0.00 \pm 0.00$ |
| **Triangle Grid** | Random-GNN | $97.80 \pm 0.91$ | $75.68 \pm 22.18$ | **$85.10 \pm 2.82$** | $66.88 \pm 13.26$ | $15.58 \pm 3.96$ | $17.09 \pm 7.88$ | $19.35 \pm 16.34$ |
| | Pretrained-GNN | $94.09 \pm 2.14$ | $0.00 \pm 0.00$ | $70.12 \pm 4.25$ | **$84.67 \pm 18.23$** | $12.98 \pm 4.24$ | $0.00 \pm 0.00$ | $0.00 \pm 0.00$ |

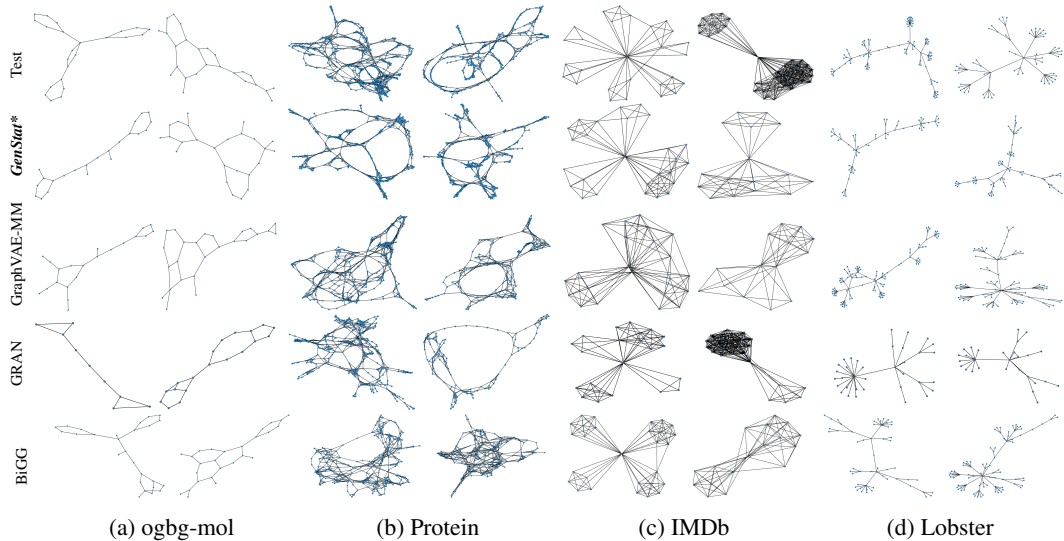

(a) ogbg-mol  (b) Protein  (c) IMDb  (d) Lobster

Figure 7: Visual comparison of *GenStat* with SOTA *adjacency-based GGMs*. The top row shows randomly selected graphs from the test set for each dataset, datasets with *varying structural characteristics*. The other rows show graphs generated by each model. The generated graphs shown are the two visually most similar samples in the generated set. The quality of generated graphs by SOTA deep GGMs is indistinguishable from those generated by *GenStat*. *Unlike adjacency-based GGMs, GenStat does not have access to nodes interaction and is restricted to certain graph-level statistics.*

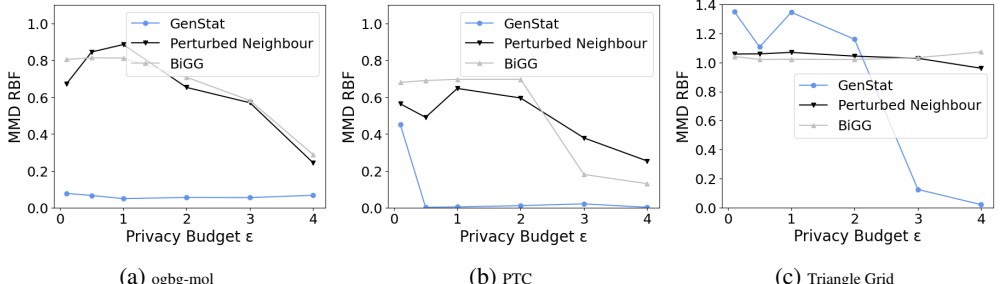

(a) ogbg-mol  (b) PTC  (c) Triangle Grid

Figure 8: Comparison of *GenStat* (statistics-based GGM) with BiGG (adjacency-based GGM) under $\epsilon$-Edge LDP guarantee, in terms of the *Random GNN-Based MMD RBF* score. A lower score is better. The lower bound $\epsilon = 0$ ensures perfect privacy. It was not feasible to train BiGG on the dense Protein dataset, due to out-of-memory Error.

Table 6: Comparison of *GenStat* with *adjacency-based* GGMs for the *statistic-based MMD* score (lower is better). The best result is in bold and the second best is underlined. Compared to the state-of-the-art GGMs (*that require access to all adjacencies*), the *GenStat* graphs reach competitive graph quality for real datasets, while limited to graph-level statistics.

| Dataset | Statistic | 50/50 split | *GenStat*\* | GraphVAE-MM | GraphRNN-S | GraphRNN | GRAN | BiGG |
|---|---|---|---|---|---|---|---|---|
| **ogbg-molbbbp** | Deg. | $2e^{-4}$ | 0.004 | **0.001** | 0.016 | _0.002_ | 0.008 | 0.003 |
| | Clus. | $2e^{-5}$ | **3e$^{-4}$** | 0.005 | 0.572 | _9e$^{-4}$_ | 0.353 | 0.001 |
| | Orbit. | $9e^{-5}$ | 0.009 | $8e^{-4}$ | 0.006 | _4e$^{-4}$_ | 0.013 | **5e$^{-5}$** |
| | Spect. | $5e^{-4}$ | 0.015 | **0.005** | 0.045 | 0.135 | 0.056 | _0.007_ |
| | Diam. | 0.002 | 0.036 | **0.018** | 0.199 | 0.495 | 0.317 | _0.033_ |
| **Protein** | Deg. | $4e^{-5}$ | _0.002_ | 0.006 | 0.046 | 0.012 | **0.003** | 0.007 |
| | Clus. | 0.004 | 0.172 | **0.059** | 0.324 | 0.123 | _0.059_ | 0.099 |
| | Orbit. | $5e^{-4}$ | _0.112_ | 0.152 | 0.316 | 0.264 | **0.053** | 0.316 |
| | Spect. | $4e^{-4}$ | 0.008 | _0.007_ | 0.028 | 0.018 | **0.004** | 0.012 |
| | Diam. | 0.003 | _0.012_ | 0.091 | 0.302 | 0.342 | **0.009** | 0.181 |
| **IMDb** | Deg. | 0.001 | 0.048 | 0.061 | 0.439 | 0.353 | **0.018** | _0.023_ |
| | Clus. | 0.004 | 0.291 | 0.303 | 1.052 | 1.051 | _0.019_ | **0.071** |
| | Orbit. | 0.004 | **0.020** | 0.082 | 1.214 | 0.935 | _0.033_ | **0.030** |
| | Spect. | 0 | 0.054 | 0.074 | 0.502 | 0.417 | **0.018** | _0.022_ |
| | Diam. | $4e^{-5}$ | 0.059 | 0.043 | 1.023 | 0.490 | **1e$^{-4}$** | **1e$^{-4}$** |
| **Mutag** | Deg. | $3e^{-4}$ | _0.001_ | _0.001_ | 0.006 | 0.006 | **6e$^{-4}$** | 0.004 |
| | Clus. | 0 | 0.004 | **0** | $5e^{-4}$ | 0.21 | 0.015 | _2e$^{-4}$_ |
| | Orbit. | $1e^{-5}$ | 0.002 | _1e$^{-4}$_ | 0.002 | $8e^{-4}$ | 0.007 | _3e$^{-4}$_ |
| | Spect. | 0.005 | 0.036 | _0.019_ | 0.105 | 0.070 | 0.053 | **0.015** |
| | Diam. | 0.013 | 0.072 | **0.015** | 1.157 | 0.819 | 0.685 | _0.073_ |
| **PTC** | Deg. | $1e^{-4}$ | 0.037 | 0.020 | 0.022 | _0.005_ | 0.013 | **0.001** |
| | Clus. | $9e^{-5}$ | 0.008 | **3e$^{-4}$** | 0.254 | 0.003 | 0.137 | 0.002 |
| | Orbit. | $8e^{-5}$ | 0.005 | 0.003 | 0.035 | _0.002_ | 0.006 | **7e$^{-5}$** |
| | Spect. | 0.002 | _0.032_ | 0.018 | 0.057 | 0.075 | 0.034 | **0.004** |
| | Diam. | 0.013 | 0.265 | _0.043_ | 0.270 | 0.397 | 0.194 | **0.010** |
| **Triangle Grid** | Deg. | $3e^{-5}$ | 0.075 | **0.001** | 0.053 | _0.033_ | 0.134 | **0.001** |
| | Clus. | 0.002 | 1.175 | **0.093** | 1.094 | 1.167 | 0.678 | _0.107_ |
| | Orbit | $8e^{-5}$ | 0.239 | **0.001** | 0.121 | 0.107 | 0.673 | _0.004_ |
| | Spect | 0.004 | 0.0361 | **0.013** | 0.033 | 0.030 | 0.184 | _0.020_ |
| | Diam. | 0.014 | _0.499_ | **0.133** | 1.124 | 1.121 | 1.133 | 1.123 |
| **Grid** | Deg. | $1e^{-5}$ | 0.044 | **5e$^{-4}$** | 0.014 | 0.013 | 0.003 | _0.002_ |
| | Clus. | **0** | **0** | **0** | 0.003 | 0.166 | 1e$^{-4}$ | _3e$^{-5}$_ |
| | Orbit. | $2e^{-5}$ | 0.015 | **0.001** | 0.090 | 0.019 | 0.007 | _0.003_ |
| | Spect. | 0.004 | 0.040 | _0.014_ | 0.112 | 0.111 | **0.012** | 0.018 |
| | Diam. | 0.014 | 0.314 | **0.065** | _0.128_ | 0.461 | 0.281 | 0.328 |
| **Lobster** | Deg. | 0.002 | 0.014 | **2e$^{-4}$** | 0.016 | 0.004 | 0.005 | _0.001_ |
| | Clus. | 0 | 0.005 | **0** | 0.319 | **0** | _0.304_ | **0** |
| | Orbit | 0.002 | 0.084 | _0.008_ | 0.285 | 0.033 | 0.331 | **6e$^{-4}$** |
| | Spect | 0.005 | 0.051 | _0.017_ | 0.045 | 0.035 | 0.043 | **0.012** |
| | Diam. | 0.032 | 0.335 | _0.187_ | 0.242 | 0.384 | 0.446 | **0.101** |

Table 7: The effectiveness of *GenStat* for benchmarking GNNs on Protein dataset. The Table shows GNN models' link prediction accuracies trained and tested on a) Original graphs (second column) and b) *GenStat* graphs (third column) and the resulting correlations/MSE benchmark effectiveness scores for the Protein dataset.

| GNN Model | Accuracy on Original graphs | Accuracy on *GenStat* graphs | Pearson | Spearman | MSE |
|---|---|---|---|---|---|
| GAT | 63.05 | 69.14 | 0.6223 | 0.6 | 0.0036 |
| GCN | 63.39 | 66.29 | | | |
| GIN | 55.33 | 65.01 | | | |
| SGC | 57.68 | 59.91 | | | |

Table 8: Comparison of the benchmark effectiveness of *GenStat* , Statistic-based GGM, with that BiGG, the SOTA adjacency-based GGM. The benchmark effectiveness of *GenStat* for link prediction is impressive, considering that the model does not observe specific links during training time (unlike BiGG).

| | Pearson | | Spearman | | MSE | |
|---|---|---|---|---|---|---|
| Dataset | BiGG | *GenStat\** | BiGG | *GenStat\** | BiGG | *GenStat\** |
| ogbg-molbbbp | **0.8069** | 0.7468 | 0.4 | **0.8** | 0.0010 | **0.0005** |
| Protein | **0.9127** | 0.6223 | **0.8** | 0.6 | 0.0187 | **0.0036** |
| IMDb | 0.0596 | **0.0873** | 0.0 | **0.4** | 0.0344 | **0.0062** |
| PTC | **0.8928** | 0.5570 | **0.8** | 0.6 | **0.0018** | 0.0038 |
| MUTAG | **0.9857** | 0.8900 | **0.8** | 0.4 | **0.0010** | 0.0014 |
| Lobster | **0.9148** | 0.5046 | **0.8** | 0.6 | **0.0026** | 0.0066 |
| Grid | 0.6896 | **0.9983** | 0.8 | **1.0** | 0.0042 | **0.0001** |
| Triangle Grid | **0.9637** | 0.8518 | **1.0** | 0.4 | 0.0158 | **0.0023** |

Table 9: Comparison of deep GGMs in terms of *train time*. The numbers show the average training time per epoch. We do not show results for GraphRNNs, it is much slower than later methods [89].

| Dataset | GRAN | BiGG | GraphVAE-MM | *GenStat\** |
|---|---|---|---|---|
| **ogbg-mol** | 4.62 | 23.14 | 2.71 | 0.721 |
| **Protein** | 9.51 | 130.28 | 4.87 | 2.59 |
| **IMDb** | 1.39 | 19.42 | 2.67 | 0.388 |
| **PTC** | 2.36 | 7.66 | 0.33 | 0.139 |
| **MUTAG** | 0.90 | 5.20 | 0.16 | 0.079 |
| **Lobster** | 1.12 | 18.02 | 0.15 | 0.041 |
| **Grid** | 7.62 | 97.75 | 0.49 | 0.14 |
| **Triangle Grid** | 12.61 | 82.86 | 0.42 | 0.07 |

Table 10: Comparison of deep GGMs in terms of *generation time*. The numbers show the average generation time per batch. Small numbers are rounded up to 0.001.

| Dataset | GRAN | BiGG | GraphVAE-MM | *GenStat\** |
|---|---|---|---|---|
| **ogbg-mol** | 37.62 | 0.11 | 0.001 | 0.001 |
| **Protein** | 44.19 | 2.93 | 0.001 | 0.001 |
| **IMDb** | 2.36 | 0.24 | 0.001 | 0.001 |
| **PTC** | 0.7 | 0.07 | 0.001 | 0.001 |
| **MUTAG** | 24.63 | 0.08 | 0.001 | 0.001 |
| **Lobster** | 1.34 | 2.00 | 0.001 | 0.001 |
| **Grid** | 29.27 | 0.31 | 0.001 | 0.001 |
| **Triangle Grid** | 22.11 | 2.59 | 0.001 | 0.001 |