# OpenReview forum: "Neural Graph Generation from Graph Statistics"
_NeurIPS.cc/2023/Conference — NeurIPS 2023 poster_

### Official Review · Reviewer_bJqH · 2023-07-06

**Soundness:** 3 good
**Presentation:** 4 excellent
**Contribution:** 3 good
**Rating:** 5
**Confidence:** 4

**Summary:**

This paper proposes a deep graph generative model from aggregate graph statistics, to protect privacy. The authors designed an inference model to learn latent variables from graph statistics. And the probabilistic model keeps permutation invariance.

**Strengths:**

1. To learn the intractable latent variables, the paper proposes a reasonable lower bound.
2. The probabilistic model could keep permutation invariance.
3. The model can learn graph statistics explicitly.

**Weaknesses:**

1. I think it is time-consuming for the descriptor to conclude the graph statistics compared with the adjacency matrix-based model, especially when the number of graphs is large/each graph is large. So the comparison with such models is somewhat unfair.

2. The performance high relies on which graph statistics are used. Without enough/reasonable statistics, the model can not capture the complex dependency.

3. The assumption for latent variables $\tilde{A}$ to be independent will hurt the expressiveness of the model.

4. The model is hard to be used on molecules. First this  model can not be used on graphs with node features. Second, it is impossible to learn the molecule structures with graph-level statistics (also very hard to design such statistics with node features).

**Questions:**

1. Why do we have to use graph statistics to protect privacy? Why not simply hide the true names/items?
2. During the inference stage, there are two probabilistic models, which take more time compared with GraphRNN. Can you give a computation complexity analysis compared with popular baselines? (GraphRNN, GRAN)

**Limitations:**

See weakness.

---

> ### Author Rebuttal · Authors · 2023-08-07
>
> We appreciate the reviewer's attention to the paper. Our responses to Weaknesses (W) and Questions (Q) are provided below; all references and citations refer to the paper.
>
> **W1.** We agree that calculating graph statistics has computational overhead. However, the training time of GenStat, *which includes time to conclude graph statistics by descriptor functions*, is substantially faster than those for deep adjacency-based GGMs, up to 2 orders of magnitude compared to SOTA autoregressive models (see Table 7 in the appendix). Evaluating the edge reconstruction probability is expensive and tends to dominate the training time of adjacency-based GGMs (lines 282-283). In addition, graph statistics can be calculated in parallel in near constant time for small and medium size graphs.
>
> **W2.** We agree that the performance of GenStat is influenced by graph statistics. However, our experimental results on 8 datasets, spanning various domains and exhibiting diverse structural properties, demonstrate that the default graph statistics employed in GenStat are capable of modelling graphs with a wide range of structural characteristics. We also should emphasize that GenStat is a flexible framework (lines 95-97), allowing for the incorporation of graph statistics of interest (which satisfy the properties discussed in Section 3.5) in a target application. O’Bray et al. [51] note that different graph statistics are important for different applications. O’Bray et al. [51] therefore advise selecting a GGM based on the graph statistics of interest in the target application.  The adaptability of GenStat enables researchers to tailor the model to suit different applications effectively.
>
> **W3.** We agree that the independence assumption may affect the model’s expressiveness. However, it resulted in much more efficient computations. Moreover, even with the independence assumption, GenStat ranks among the top two models in 8 out of 10 reported MMD RBF scores, for real-world datasets (see Table 2), in comparison with *SOTA adjacency-based GGMs*.
>
> **W4.** *Our experiments actually provide evidence that it is possible to learn molecular structures with graph-level statistics.* We evaluate GenStat on ogbg-molbbbp (*a molecule dataset* containing 2039 real-world molecular graphs) as well as MUTAG and PTC (datasets of chemical compounds). In comparison with deep GGMs, trained on complete adjacency matrices, GenStat is competitive in terms of both GNN-based and statistics-based evaluation metrics (see Tables 2 and 5). In particular, GenStat outperforms the SOTA adjacency-based deep GGMs on ogbg-molbbp (molecular dataset) in terms of GNN-based evaluation metrics (Table 2). We achieved these results with our default statistics. An interesting direction for future research would be to consider domain-specific statistics for molecular graph generation. A possible candidate for such statistics could be derived from the semantic constraints introduced in previous work on molecule graph generation [42].
>
> The model can expand to generate the node and edge features. For example, as in GraphVAE [66], the decoder generates the node and edge features. The graph descriptors will be defined as functions of both feature and edge matrices, see lines 295 - 297. Our paper follows the graph structure learning research track and shows that modelling aggregated graph statistics displays competitive quality to SOTA adjacency-based GGMs while guaranteeing ε-Edge LDP. SOTA deep GGMs [6, 82, 83] and other traditional  GGMs (BTER, Chung-Lu) that focus on the problem of learning structural information from graphs do not accommodate node and edge features and we wanted to use exactly their setting. We agree that attributed graphs are an important topic and discuss extending statistics-based graph generation to attributed graphs as a valuable direction for future work (lines 321-325).
>
> **Q1.** *Graph anonymization techniques* enable the public release of private graphs by hiding the true names/items. These techniques have two main limitations, as discussed in previous works [47] and https://ieeexplore.ieee.org/stamp/stamp.jsp?tp=&arnumber=10098897&tag=1
> 1) These techniques provide limited protection, typically only against specific known attacks.
> 2) These techniques are mainly applicable in a centralized setting with a trusted data curator.
>
> As we discussed in the paper, Statistics-based graph generation supports the more challenging use case of decentralized graphs where privacy concerns rule out collecting raw data in a central repository (see lines 27-37). In addition, as we proved in  Lemma 1. of Proposition 2 (lines 578-584) collected perturbed graph statistics satisfy Edge LDP and are secure against post-processing attacks.
>
> **Q2.** Using two probabilistic models can cause computation overhead. However, compared to autoregressive models, GenStat benefits from parallel computation, leading to faster training and generation times, up to 2  orders of magnitude (see Tables 7 and 8).
>
> The Table below compares the computational complexity of popular GGMs; N and \|E\| are nodes and edges numbers. Note for GenStat, we assumed matrix multiplication with O(N^3)  complexity to exactly compute the descriptors. Approximating graph statistics [16, 32, 55] and exploiting the sparsity of real-world graphs can substantially reduce the computational cost (see lines 298-300).
> | Model| Auto-Regressive Steps| Train  | Generation|
> |-|-|-|-|
> | GenStat| O(1) | O(N^3) | O(N^2)|
> | BiGG [6] |log(N)| O(min((\|E\|+N)log(N), N^2)) | O(min((\|E\|+N)log(N), N^2))|
> | GRAN [4] |N| O(N^2)   | O(N^2)|
> | GraphRNN [82]| \|E\| * N| O(N^2)| O(N^2)|
>
> Autoregressive models generate the graphs sequentially, an element at each step where each element is conditioned on the previous ones. The dependencies between steps, make it difficult to parallelize the training process. GenStat generates a graph (updates the gradients) in parallel, which leads to substantially faster training.

---

### Official Review · Reviewer_hsws · 2023-07-06

**Soundness:** 2 fair
**Presentation:** 3 good
**Contribution:** 2 fair
**Rating:** 4
**Confidence:** 3

**Summary:**

This paper considers a new setting that learns a deep graph generative model from aggregate graph statistics rather than graph adjacency matrix. The motivation for generating graphs from statistics is privacy preservation. For this task, the authors propose a simple VAE model, where the encoder takes graph statistics as input and the decoder outputs graphs that have similar statistics with the input. Experiments demonstrate the effectiveness of the proposed method to some extent.

**Strengths:**

1.	This paper proposes a new setting, i.e., learning a deep model for statistics-based graph generation.
2.	The proposed model generates graphs with more similar statistics with training sets than previous parametric models.

**Weaknesses:**

1.	My main concern is about the task setting.
a)	What's the application of the generated graphs. In another word, what downstream tasks can the generated graph be used for?
b)	Which datasets support the claimed application scenario?
c)	The evaluation metrics are not straightforward enough. To demonstrate the usefulness of the generated graphs, I think it necessary to provide evaluation results on downstream tasks.
d)	If only easy statistics are preserved, how to make sure that the generated graphs benefit downstream tasks?
e)	How are the considered graph statistics determined?
2.	The technical contribution is minor, because the proposed model is a naive invariant of VAE. Otherwise, the authors may want to highlight the technical difficulties.
3.	Some typos. Line 47 ‘(Appendix Sec. 7.11)’: There is no Sec. 7.11. Line 87: Maybe you forget to bolden Federated Learning. Line 245 ‘87.37% vs. 23.76 for Pretrained-GNN’: Maybe you wrongly copy the results. Same mistake in Table 3.

**Questions:**

See Weakness 1.

---

> ### Author Rebuttal · Authors · 2023-08-08
>
> We appreciate the reviewer's attention to the paper. Our responses to Weaknesses (W) are provided below; all references and citations refer to the paper.
>
> **W1.a and W1.b. Task Setting and Application of Generated Graphs.** As we explain in related work (lines 55-61 and Figure 5), we start with *realistic graph generation* as our basic task that has been studied extensively in previous work for decades, both in parametric and neural approaches, including many papers published at NeurIPS (Zahirnia et al. (2022) and Liao et al. (2019)) and ICML ( Dai et al. (2020); You et al. (2018)) which are our main baselines. The research question we address is how to learn to generate realistic graphs while protecting private information from raw data.\
> One of the lessons from the recent impact of generative AI is that generative models are useful independent of downstream tasks. For example, many research groups and companies have addressed the task of generating realistic images, independent of downstream tasks such as image classification. Experimental settings, datasets, and evaluation metrics used in this study closely follow recent studies in realistic GGMs You et al. (2018); Zahirnia et al. (2022); Dai et al. (2020).
>
> **W1.c. Complex Evaluation Metrics and Downstream Tasks for Models Evaluation.** The metrics we employ in this paper for assessing the quality of generated graphs are not straightforward because quantifying the realism of generated structured objects is not straightforward. Indeed, there are research papers devoted to the question of how to evaluate the quality of generated objects, both for images (https://arxiv.org/pdf/2206.10935.pdf) and graphs O’Bray et al. (2022); Shirzad et al. (2022). We use the SOTA graph quality metrics proposed in the previous literature. Lines 215-228 and Appendix 7.6 explain these metrics.\
> While it is true that generative models support downstream applications, the field of generative graph modelling has in general moved towards more direct measures of generated graph quality. Two main reasons are 1) Performance on downstream tasks may not correlate with the quality of generated objects. For example in graph classification, SOTA performance is achieved by differentiable pooling methods, that transform input graphs into sparse unrealistic versions before embedding them for classification. 2) There are potentially many downstream tasks (e.g. node classification, link prediction) and many datasets for evaluating them. It is not clear which tasks/datasets to pick to obtain a ranking of graph generative methods. For further discussion, please see the references we cite, especially O’Bray et al. (2022).
>
> **W1.d. Benefit of Generated Graphs, with Limited Statistics, for Downstream Task.** Like other areas of generative AI, graph generative modelling has moved towards directly estimating the realism of the generated objects. The GNN-based evaluation metrics, used in this and previous studies, are independent of graph statistics or specific downstream tasks and can compute the dissimilarity between any two sets of graphs regardless of the domain, as discussed in (O’Bray et al., 2022). The GNN-based representation allows us to evaluate the model on downstream tasks that were not known during the learning of the representation, as discussed in Shirzad et al. (2022). The statistics we investigate in this paper are good default statistics that are effective across a variety of domains. As suggested in Zahirnia et al. (2022), for specific domains/tasks they can be augmented with domain-specific statistics in a straightforward and flexible manner.
>
> **W1.e. Determining the Graph Statistics.** The default graph statistics used in GenStat are 1) permutation-invariant; 2) aggregations of node-level statistics (that can support Edge LDP); and 3) differentiable (see lines 166- 173). These statistics are known from prior research to be generally important for graph modelling across different domains and are easy to interpret. We should emphasize that GenStat is a flexible framework (lines 95-97), allowing for the incorporation of any graph statistics of interest which satisfy the properties discussed in Section 3.5 (lines 166-173).
>
> **W2. Technical Contribution Compared to a Simple VAE.** While GenStat relies primarily on VAE, as opposed to standard Graph VAE, it is designed to only depend on integrated observed graph-level statistics, which is not a naive extension to VAE.
> - This work, compared to simple VAE, proposes a novel two-level probabilistic model in which the adjacency matrix is latent (as well as the graph-level representation Z) and observed variables are limited to a predefined set of graphs’ statistics (Figure 2). Under this setting, the marginal distribution over the graph statistics is a double integral. We approximate the mixture integral with a Variational Auto-Encoder (VAE) and Monte Carlo decoder.
> - To make the graph-level statistics differentiable and informative, we adopt adaptive soft histogram functions to summarize graphs in permutation invariant vectors with a dimensionality of O(log(N)). For each graph invariant, the diagonal co-variance parameter is computed using the maximum likelihood estimate given the estimated means.
>
> **W3. Typos**
> - **Q.** Line 47 ‘(Appendix Sec. 7.11)’: There is no Sec. 7.11.\
> **A.** Thank you for identifying the typo. We have now updated the text to refer to (Appendix Sec. 7.9).
> - **Q.** Line 87: Maybe you forget to bolden Federated Learning.\
> **A.** Thank you for identifying the possibility of a typo. We intended to study Federated Learning under the "Graphs and privacy." subsection as a graph privacy-preserving paradigm. We have not made any changes to the text.
> - **Q.** Line 245 ‘87.37 vs. 23.76 for Pretrained-GNN’: Maybe you wrongly copy the results. Same mistake in Table 3.\
> **A.** Thank you for identifying the typo. The mentioned results are for the Protein dataset, not IMDb. We have now updated the text.

---

> > ### Comment · Reviewer_hsws · 2023-08-15
> > **Thanks for the response.**
> >
> > I appreciate the authors for their efforts in responding to our questions. However, my concerns have not been properly addressd.
> >
> > 1. Task setting and application. Though realistic graph generation is a common topic, this paper considers generating graphs only from statistics, which is different from the several literatures given in the author rebuttal. Previous graph generation techniques can be applied in many real scenarios such as drug discovery. However, the potentioal application of the new setting and the proposed method are not well demonstrated.
> >
> > 2. I still think downstream task is important. In my understanding the graphs generated by the propsoed method under the task setting should be used for research, as a replacement when real graphs are not available to the researchers. I am not sure whether the graphs can work well in this task. If no specific downstream task is conduct, the application potential of the method is limited.
> >
> > 3. I am not very convinced with the technical difficulty of the proposed method compared with standard VAE. But this is not the main reason why I recommand for reject. I would raise my score if my above two concerns are well addressed.

---

> > > ### Author Response · Authors · 2023-08-18
> > > **Evaluation On Downstream Task and GenStat Application**
> > >
> > > We appreciate the time taken for the review and the comments provided in the rebuttal. In this response, we aim to further address the concerns raised by the reviewers through the following actions:
> > >
> > > **1- Applications in General.** Our framework supports training any graph generative model class, and therefore supports any application of graph generative modelling. The difference is that we use different information during the training process than what deep graph models are traditionally trained on (graph statistics vs. completed adjacency matrix). For the drug discovery domain that you mentioned, our experiments on MUTAG and PTC provide evidence that GenStat can generate realistic chemical compounds; see Table 2 in the paper. As we mentioned under Future Work, evaluating GenStat on task-specific metrics like novelty and validity [42] is an important direction.
> > >
> > > **2- GGMs for Benchmarking GNNs.** A major application of graph generation is to provide the research community with synthetic graphs for benchmarking the performance of graph neural networks (GNNs) on downstream tasks (e.g., link prediction). A prominent recent example is the paper "Graph Generative Model for Benchmarking Graph Neural Networks", published by the Google graph research team at ICML 2023 (our reference 81; Yoon et al.). They note that the number and variety of real-world graph datasets for benchmarking GNNs is limited, and should be complemented with synthetic graphs. They agree with us that synthetic graph generation can address privacy concerns and should produce realistic graphs. In addition, they evaluate the “benchmark effectiveness" of synthetic graph
> > > generation for ranking GNNs on the downstream task of link prediction. Following their methodology,
> > > we evaluate the effectiveness of GenStat for benchmarking GNNs on link prediction. *We find that it is highly effective—thank you for suggesting that we include this kind of metric related to a downstream task*. The details are as follows.
> > >
> > > The idea behind benchmark effectiveness is that “performance rankings among *m* GNN models on generated graphs should be similar to the rankings among the same *m* GNN models on the original graphs" [81]. The methodology of [81] is to evaluate each GNN model on the downstream task twice: First on the original dataset, second on a synthetic dataset generated by the GGM. The benchmark effectiveness of the GGM is then measured by the correlation between the GNNs task scores from the original and the task scores from the synthetic data
> > >
> > > For example, for link prediction, the procedure is as follows.
> > > 1. Train a GGM (e.g., GenStat) on the original dataset (e.g., IMDb). Generate a synthetic graph dataset of the same size as the original one (e.g. IMDb-Synth).
> > >  2. Score each GNN model (e.g., GCN) on the original graph dataset for the downstream task. E.g. for link prediction, divide links into training and test edges, train the GNN, report test performance (e.g. Accuracy is the GNN test metric in [81]). Call this score GNN-original.
> > > 3. Score each GNN model (e.g., GCN) on the synthetic graph dataset for the downstream task. Call this score GNN-synthetic.
> > > 4. For each dataset, report the correlation between the GNN-original and the GNN-synthetic scores. Yoon et al. [81] use Pearson correlation, Spearman rank correlation, and MSE (squared difference beween the accuracy scores).
> > >
> > > Table 1 illustrates the idea by showing the link prediction accuracies and the resulting correlations/MSE benchmark effectiveness scores for the IMDb dataset.
> > >
> > > Table 1. The effectiveness of GenStat for benchmarking GNNs on IMDb.
> > > |GNN Model|Accuracy on Original graphs|Accuracy on GenStat graphs|Pearson|Spearman|MSE|
> > > |-|-|-|-|-|-|
> > > |GCN|0.67|0.61|0.42|0.4|0.002|
> > > |GIN|0.66|0.67|
> > > |SGC|0.70|0.69|
> > > |GAT|0.68|0.61|
> > >
> > > Table 2 compares the benchmark effectiveness of GenStat with that BiGG, the SOTA adjacency-based
> > > GGM. As we noted in the main paper, this is an apples-to-oranges comparison because BiGG has
> > > access to the entire adjacency matrix. We use the datasets discussed in section 4: 8 graph datasets
> > > that include synthetic and real world graphs. Following [81] we choose popular GNN models for
> > > benchmarking: GCN (Kipf & Welling, 2016), GIN (Xu et al., 2018), SGC (Wu et al., 2019), and
> > > GAT (Velickovic et al., 2017). Link prediction was performed with a dot product decoder as in
> > > [81].
> > >
> > > Table 2. The effectiveness of GGMs for Benchmarking GNNs on link prediction.
> > > |||GenStat (Ours)|||BiGG||
> > > |-|-|-|-|-|-|-|
> > > |Dataset|Pearson|Spearman|MSE|Pearson|Spearman|MSE|
> > > |IMDb|**0.42**|**0.4**|0.0021|0.11|-0.4|0.0008|
> > > |Protein|**0.65**|**0.4**|0.002|0.40|**0.4**|0.0022|
> > > |MUTAG|**0.96**|**0.8**|0.0004|0.85|0.2|0.0006|
> > > |PTC|0.49|0.4|0.0047|**0.96**|**0.8**|0.0002|
> > > |Grid|0.57|0.4|0.0025|**0.99**|**1**|0.0001|
> > > |lobster|**0.97**|**1**|0.0003|0.95|**1**|0.0001|
> > > |ogbg-mol|0.95|**1**|0.0003|**0.98**|0.8|0.0001|
> > > |Triangle Grid|**0.70**|**0.4**|0.0042|0.60|**0.4**|0.0005|
> > >
> > > *Please refer to the following comment as well.*

---

> > > > ### Author Response · Authors · 2023-08-18
> > > >
> > > > **Results:** *On 6 out of 8 datasets, the GenStat benchmark effectiveness is competitive with or superior to the benchmark effectiveness of BiGG.* On the IMDb dataset, GenStat shows 0.30 higher Pearson and .80 higher Spearman correlations than BiGG. For two datasets, PTC and Grid (with highly regular local structure), the benchmark effectiveness of BiGG is much better, but the GenStat graphs still show a substantive correlation (at least 0.4985 in the Pearson correlations). For both GGM models the MSE is small. In our opinion, *the benchmark effectiveness of GenStat for link prediction is impressive, considering that the model does not observe specific links during training time (unlike BiGG)*. We note that while the study of [81] used different datasets, the highest Pearson correlation reported by the GGMs they studied was 0.88, whereas Genstat achieves above 0.9 in 3 out of 8 datasets.
> > > >
> > > > **Reproducibility:** The code and data used in this benchmark effectiveness experiment is added to our anonymous repository, https://github.com/ddccbbee/GenStat. The GNNs.py contain the GNNs implementation. LinkPredictionFramewok.py contain the main training pipeline. ReportedResult directory contains the reported result.

---

> > > > > ### Comment · Reviewer_hsws · 2023-08-19
> > > > > **Thanks for the response. My concerns are not fully addressed.**
> > > > >
> > > > > I appreciate the authors' efforts in improving this work. However, my concerns are not fully addressed.
> > > > >
> > > > > What I challenge indeed is the setting of generating graphs only from statistics. I think the authors may want to find a specific scenario to demonstrate the significance of this setting, and conduct experiments on datasets from this scenario. It seems that currently considered tasks are those where we can obtain the adjacency matrix. Therefore, I am not convinced with the importance of developing the technique of generating graphs only from statistics.
> > > > >
> > > > > Though I admit that the downstream task in Table 1 shows the effectiveness of the method, I do not think it has well demonstrated the usefulness of the proposed statistics based method, because it seems that in the task there is no need of only using statistics. Also, I have an additional question. Since BiGG uses the adjacency matrix, while GenStat only uses statistics information, why does GenStat outperforms BiGG in Table 2? I doubt whether BiGG is well tuned.

---

> > > > > > ### Author Response · Authors · 2023-08-20
> > > > > >
> > > > > > We appreciate the reviewer’s positive feedback, especially regarding their observation that "the downstream task in Table 1 shows the effectiveness of the method."
> > > > > >
> > > > > > However, we respectfully disagree with the reviewer that in "benchmarking effectiveness of GNNs" there is always access to adjacency matrixes. The motivation for using GGMs for benchmarking GNNs is privacy when the access to graph is limited regarding the sensitivity of information [81]. *Benchmarking GNNs on distributed graphs with sensitive edge information is one of the settings where there is no access to the adjacency matrix;* examples of these graphs are 1) distributed social network, e.g. Mastodon, or 2) social graphs, e.g users’ connections through their phone contact lists, face-to-face interaction, sexual and friendship networks [51].  In this setting, GenStat can be used to generate synthetic graphs which enable GNNs benchmarking. We provide an example in the introduction (lines 31-34) of how Google, Apple and Microsoft collect perturbed a collect node-level graph statistics since there is no access to the actual adjacency matrix regarding the users’ privacy. In the Google example, the collected graph statistics can be used to generate synthetic graphs which then enable benchmarking of GNNs on the User-Website Interaction data. We compared GenStat with the adjacency-based GGM model to demonstrate the effectiveness of our model and showed even with much-limited input information, Genstat can outperform SOTA adjacency-based GGMs.
> > > > > >
> > > > > > **Advantages of GenStat (statistic-based GGM) in general.** GenStat, when compared to adjacency-based GGMs, offers the following advantages:
> > > > > > - GenStat is permutation invariant independent of the utilized neural network architecture, observation 1 (lines 142-147)
> > > > > > - GenStat is substantially faster than GGM benchmarks, up to two orders of magnitude in train and generation time, see Tables 7 and 8.
> > > > > > - GenStat is the first deep graph generator that "does not require observations of individual nodes or edges" and guarantees Edge LDP.
> > > > > > -  Although GenStat is limited to simple graph statistics, in our experiments for real-world datasets, it almost always ranks among the top two models in all of the empirical experiments, see Table 2.
> > > > > >
> > > > > > We believe with all of these advantages the proposed modelling graph statistics approach, as also
> > > > > > mentioned by reviewer 92Y3, is "likely to impact graph learning beyond the specific application(graph generation) presented in the paper".
> > > > > >
> > > > > > **BiGG.** We have used the hyperparameters and the original implementation released by Dai et al. (2020). Our experimental outcomes, for all the datasets, closely align with the results documented in the original work by Dai et al. (2020). This congruence serves as additional confirmation of the model's successful training. Samples generated for each dataset are accessible on our anonymous Google Drive repository https://drive.google.com/drive/folders/1mF-kU021-ceNh6ejLgf41sSE9FEzc01Q
> > > > > >
> > > > > >
> > > > > > **Effectiveness of GenStat in GNN benchmarking.** Tables 1, 2, 3, and 5 illustrate the effectiveness of graphs generated by GenStat, showcasing its competitiveness or superior performance when compared to BiGG and adjacency-based GGMs on real datasets in terms of GNN-based evaluation methods [64, 51]. the GNN-based evaluation metrics are not task-specific and can compute the dissimilarity between any two sets of graphs regardless of the domain and a specific task [51].In fact, the representation of generated graphs by GenStat closely resembles the original data. Consequently, a high correlation in GNN model performance between generated and original graphs can be anticipated.
> > > > > >
> > > > > > **Specific scenario which demonstrates the significance of statistic-based learning.**  In section "4.3" of the paper, we conduct a focused analysis on graph generation when access to the row adjacency matrix for preserving edges' privacy (Edge-LDP) is unavailable. In this setting the nodes' neighbour list (rows of adjacency matrix) can not be collected in a trusted data collector. We compare GenStat against Edge-LDP versions of adjacency-based deep GGMs (Figure 4).   Our experiment shows that GenStat generates realistic graphs for all privacy budgets, as indicated by much lower MMD-RBF scores, and is substantially less sensitive to the imposition of stricter privacy requirements (272-280), see Figure 4.

---

> > > > > > > ### Comment · Reviewer_hsws · 2023-08-21
> > > > > > > **Thanks for the response. Increasing my score from 3 to 4.**
> > > > > > >
> > > > > > > I appreciate the authors' efforts in responding to my concerns and improveing this work. I have increased my score from 3 to 4, because the authors (1) demonstrate the significance of the task setting, and (2) conduct downstream tasks to show the effectiveness of the proposed method.
> > > > > > >
> > > > > > > However, I still think the additional results have not been sufficient enough to support the claimed applications. Moreover, since the authors highlighted the literature "Graph Generative Model for Benchmarking Graph Neural Networks" in rebuttal, I read this paper and find that the setting of generating graphs from statistics in a privacy-controlled way is not a very novel one. Though this improves the soundness, it may harm the novelty to some degree.

---

> > > > > > > > ### Author Response · Authors · 2023-08-21
> > > > > > > >
> > > > > > > > We appreciate the reviewer's positive feedback and increasing the scores.
> > > > > > > >
> > > > > > > > We should emphasize that while Yoon et al. [81]  leverage DP to enforce privacy constraints on deep GGMs (lines 80-81), Genstat is the first deep GGM that guarantees Edge-LDP, a stricter privacy guarantee. In addition, while graph statistics has been utilized to improve the expressiveness of GNNs, GenStat is the first GNN model that does not require observations of adjacency matrixes and only utilizes graph statistics.

---

### Official Review · Reviewer_ijbs · 2023-07-10

**Soundness:** 2 fair
**Presentation:** 2 fair
**Contribution:** 2 fair
**Rating:** 5
**Confidence:** 4

**Summary:**

The paper proposes a new approach to learning deep graph generative models that can generate more realistic graphs while protecting local privacy. The proposed GenStat architecture uses graph statistics instead of the entire adjacency matrix to train deep generative models.

**Strengths:**

- The paper presents a new perspective to learning deep graph generative models that can generate more realistic graphs while protecting local privacy.
- The authors demonstrate that GenStat outperforms other graph generative models in terms of imitating graph statistics from training sets.

**Weaknesses:**

- The problem is not well motivated. It is unclear to me why the standard graph generation methods (trained with reconstruction loss) will have privacy issues. In practice, when the training set consists of multiple graphs (multiple molecules), a trained graph generative model can unlikely memorize the training data (since GNNs are usually not powerful enough to memorize the raw data).
- The theoretical results are limited. Although the authors show that the proposed algorithm is differentially private, it does not develop the theoretical properties of a standard graph generation and argues that those methods are not differentially private.
- The comparison with baselines is limited. The paper should focus more on comparing with modern learning based graph generators. Figure 3 and Table 1 take almost 1 page, but it is not surprising - there have been hundreds of research papers showing that learning-based graph generative models can outperform classic graph generative models. I would highly suggest the authors replace the baselines in Figure 3 with learning based graph generators.
- The description of the proposed method is vague. After reading the paper, I still could not see the exact algorithm of how a graph is generated (e.g., when nodes have attributes) with GenStat. Many implementation details are missing.

**Questions:**

- What are the limitations of graph statistics that can be used with GenStat? For example, we can view an adjacency matrix as a type of graph statistics as well.

**Limitations:**

Some of the limitations of the paper have been discussed, but it is not sufficient (see "Questions").

---

> ### Author Rebuttal · Authors · 2023-08-09
>
> We appreciate the reviewer's attention. Our responses to Weaknesses (W) and Questions (Q) are provided below; all references and citations refer to the paper.
>
> **W1.Privacy Issues in Standard GGMs:** We provide a clear motivation for the problem in the introduction (lines 23-38). Graph Generative Models (GGMs) trained with reconstruction loss, have privacy issues because they require access to the adjacency matrix to do training. An adjacency matrix contains sensitive information because it directly reveals the individuals/entities with which a person (node) has interacted. We provide an example in the introduction (lines 31-34) of how Google, Apple and Microsoft collect perturbed node-level graph statistics rather than the actual adjacency matrix in order to preserve the privacy of their users. There is a huge difference between knowing exactly which 10 websites a user visited and knowing that a user visited 10 websites. *The issue of privacy does not revolve around the GNN memorizing the raw data;* it is the collection, storage and processing of the raw data that is the privacy problem.
>
> **W2. Theoretical Results and Edge-LDP in Standard GGMs**. We consider that the paper provides two important theoretical results.
> 1) We establish that the proposed method satisfies ε-Edge Local Differential Privacy (*Edge-LDP*); (Proposition 2; lines 162-164);
> 2) We show that the gradient updates, model distribution, and inference distribution of the graph generation procedure are permutation-invariant if the descriptor functions are permutation-invariant. (Observation 1; lines 142-147).
>
> These are the two critical properties of our proposed graph generation scheme – it achieves the desired privacy while respecting the permutation invariance.
>
> With regard to *“standard”* GGMs, we consider two groups:
> - Methods that require access to adjacency matrices [6,83,40, 82] (lines 58-60) and clearly do not satisfy Edge-LDP because they explicitly process data that contains the edge information. In fact, "Edge-LDP [54,73], guarantees plausible deniability for the inclusion or removal of a particular edge associated with a node", lines 150-152. Providing adjacency-based GMMs with nodes’ interactions (adjacency matrix) removes the possibility of any deniability for nodes.
> - Methods that process graph statistics [65,5](lines 55-58). In general, these models can be used to satisfy LDP, but they are not neural methods and their performance is dramatically inferior (see Tables 1,2,4 and Figure 3).
>
> **W3. Baselines.** We remind the reviewer that the key difference between our new GenStat and the previous deep Graph Generative Models(GGM) is that the previous methods require access to potentially sensitive raw data—the entire adjacency matrix—whereas GenStat training is based on summary statistics only. See also Related Work, lines 55-66. The previous work showing that learning-based GGMs can outperform classic GGMs is not an apples-to-apples comparison because the deep GGMs use more information than the classic GGMs (again, the entire adjacency matrix rather than graph statistics).
> 1) Our comparison with classic GGMs baselines (Table 1) is, therefore, a new apples-to-apples comparison of our new deep statistics-based method with classic statistics-based methods.
> 2) In an apples-to-oranges comparison, we benchmarked our statistics-based GenStat against *5 SOTA deep GGMs (BiGG [6], GraphVAE-MM [83], GRAN [40] and GraphRNNs [82])* that use the entire adjacency matrices (Table 2).
> 3) We also compare GenStat against Edge-LDP versions of deep GGMs (Figure 4).
>
> For all models, we use SOTA evaluation metrics to compare the distance between the distribution of test and generated graphs. We use a)GNN-based [64,68] and b)statistics-based [82] evaluation metrics and c)we also compare the generated graphs by visual inspection (see Tables 1, 2, 3, 4, 5, 6 and figures 7,3 in the Appendix and Paper).
>
> As per the reviewer's recommendation, we have relocated Figure 7 from the Appendix to the Main Paper and replaced it with Figure 3.
>
> **W4. Method Description.** Node attributes: we discussed in Section 5 (lines 294-297) and following the baselines [6, 82, 83], this work focuses on simple unattributed graphs (graphs without node and edge attributes). However, GenStat can be extended to attributed/heterogeneous graphs, where nodes/edges possess attributes, including potentially sensitive ones, by defining the graph descriptors as functions of both feature matrices and edge tensors (lines 296-297).\
> Graph Generation: To generate graphs at test time, we use a standard sampling method Ma et al. [42], as follows. We sample graph embeddings from the prior p(Z), then use the decoder to compute a soft adjacency matrix Ã with entries in [0,1] representing link probabilities. We use a 0.5 threshold to convert link probabilities to hard binary links.
>
> We believe we have clearly described how GenStat is trained through the Paper and Appendix.
> - The proposed probabilistic model is explained in Sec. 3.1 (lines 108-111) as well as in Figure 2.a.
> - The objective function is discussed in Sec. 3.2 (Equation 5).
> - The model architecture is described in Sec. 7.2 (lines 591-610 ) as well as in Figure 2.b.
> - Graph statistics used in GenStat explained in Sec. 3.5 (lines 166-190).
> - The hyperparameters are discussed in Sec. 7.2 (lines 591-610 ).
> - In addition, our repository contains the implementation of the model training procedure, graph statistics, neural model as well as datasets and used hyperparameters.
>
> **Q.** We discussed the limitation of graph statistics used by GenStat in Section 3.5 (lines 166-173). The graph statistics used in GenStat are 1) permutation-invariant; 2) aggregation of node-level statistics (Eq 8); and 3) differentiable.
> *The adjacency matrix* does not have the aforementioned properties; it depends on the node ordering and is not the aggregation of node-level statistics. Consequently, it cannot be used as a statistic for GenStat.

---

> > ### Comment · Reviewer_ijbs · 2023-08-21
> >
> > Thank you for the detailed reply. I still hold the following concerns for the paper.
> >
> > > W1: An adjacency matrix contains sensitive information because it directly reveals the individuals/entities with which a person (node) has interacted
> > > This work focuses on simple unattributed graphs (graphs without node and edge attributes)
> > Wouldn't properly anonymizing the nodes, for example, removing/anonymizing any Personally Identifiable Information, solve the problem? In the particular setting of this paper where there are no node or edge features, I'm not sure why privacy would be a concern.
> >
> > > they explicitly process data that contains the edge information
> > > potentially sensitive raw data—the entire adjacency matrix
> > The authors are arguing that there is a key difference between accessing the adjacency matrix and not. I still do not get the argument. The adjacency matrix is a binary matrix. After removing features, why does accessing the adjacency matrix alone become an issue?
> >
> >
> > Thanks for moving the results from the appendix to the main paper. But I still found it unnecessary that Figure 3 and Table 1 take almost a page. The results are not surprising since the traditional graph generative models simply do not have the capacity to model the example graphs. They should be trimmed or put into the appendix.
> > Meanwhile, more discussions between the proposed methods and standard GGMs should be added, as it is still unclear why a user would prefer the proposed approach over the standard GGMs. Maybe you could add a runtime complexity comparison.
> >
> >
> > Overall, I understand all the arguments from the authors, but I would like to remain my evaluation, given my concerns above.

---

> > > ### Author Response · Authors · 2023-08-21
> > >
> > > We appreciate the time taken for the review and the comments provided in the rebuttal. In this response, we aim to further address the concerns raised by the reviewers through the following answers:
> > >
> > > **Why does accessing the adjacency matrix alone become an issue?**
> > > Releasing the binary adjacency matrix, *omitting both nodes' identifiers and edges/nodes features*, does not provide an Edge-LDP guarantee or adequately address privacy concerns. Specifically, the work by [47] demonstrates a de-anonymization algorithm purely based on the network topology can effectively reidentify the nodes in real word graphs, Twitter and Flicker.
> > >
> > >
> > > **Anomaliztion.** Graph anonymization techniques enable the public release of private graphs by hiding the true names/items. These techniques have two main limitations, as discussed in previous works [47] and https://ieeexplore.ieee.org/stamp/stamp.jsp?tp=&arnumber=10098897&tag=1
> > > - These techniques provide limited protection, typically only against specific known attacks.
> > > - These techniques are mainly applicable in a *centralized setting* with a trusted data curator.
> > >
> > > As we discussed in the paper, Statistics-based graph generation supports the more challenging use case of decentralized graphs where privacy concerns rule out collecting raw data in a central repository (see lines 27-37). In addition, as we proved in Lemma 1. of Proposition 2 (lines 578-584) collected perturbed graph statistics satisfy Edge LDP and are secure against post-processing attacks.
> > >
> > >
> > > **Advantages of GenStat (statistic-based GGM) in general.** The paper discusses the advantages of GenStat when compared to standard Deep GGMs:
> > >
> > > - GenStat is permutation invariant independent of the utilized neural network architecture, observation 1 (lines 142-147).
> > > - GenStat is substantially faster than GGM benchmarks, up to two orders of magnitude in training and generation time, see Tables 7 and 8.
> > > - GenStat is the first deep graph generator that "does not require observations of individual nodes or edges" and guarantees Edge LDP, Proposition 2 (lines 162-164).
> > > - Although GenStat is limited to simple graph statistics, in our experiments for real-world datasets, it almost always ranks among the top two models in all of the empirical experiments, see Table 2.
> > >
> > > We believe with all of these advantages the proposed modelling graph statistics approach, as also mentioned by reviewer 92Y3, is "likely to impact graph learning beyond the specific application(graph generation) presented in the paper".
> > >
> > > As per the reviewer's recommendation, we have relocated Tables 7, and 8 (Comparison of deep GGMs in terms of training and generation time) from the Appendix to the Main Paper and replaced them with Table 1.

---

> > > > ### Comment · Reviewer_ijbs · 2023-08-21
> > > >
> > > > Thank you for the prompt reply. I could understand the claims from the paper better based on the additional responses. I also appreciate the authors taking my advice regarding reorganizing the results. While I still hold my opinion that the application of the proposed method is limited, I'd like to increase my score from 4 to 5.

---

### Official Review · Reviewer_92Y3 · 2023-07-29

**Soundness:** 4 excellent
**Presentation:** 4 excellent
**Contribution:** 4 excellent
**Rating:** 7
**Confidence:** 4

**Summary:**

The authors introduce a deep graph generator model based on graph statistics. Instead of encoding the graph itself, deterministic high-level graph statistics are first computed from the graph, and these are passed to a neural encoder and graph decoder. The stated main motivation for this approach is for increased privacy: allowing graphs to be generated purely from the graph statistics may be particularly good at preserving graph-level properties while removing user-level private edge/node characteristics. As stated by the authors, it is also the first graph generator that "does not require observations of individual nodes or edges".

The authors do the following in the rest of the work:
 - define an explicit probabalistic model for graph statistics, and use it to derive a variational lower bound for the graph encoding problem.
 - define a permutation-invariant inference distribution for their encoder ("GenStat") which can be used for link prediction / graph classification
 - analyze the privacy properties of GenStat
 - benchmark GenStat on standard graph generation evaluations
 - provide a privacy/accuracy tradeoff analysis of GenStat and the most competitive baseline

**Strengths:**

To the best of my reading, the manuscript appears quite technically sound, with a well-defined problem statement, sound probabalistic setup, accurate loss derivations, and appropriate theoretical analyses. The empirical study is thorough and appropriate for the problem. The writing is clear and easy to follow.

The contribution itself is interesting and likely to impact graph learning beyond the specific application (graph generation) presented in the paper. The joint distribution of graphs and their derived statistics is an interesting topic of study that has been explored independently in various areas of graph ML.

**Weaknesses:**

I identify one main weakness, which contributed to my score of 3 and not 4 in the "Soundness" category. Based on Figure 4, it seems as though the proposed method GenStat does not respond to the privacy budget: in other words, the measurement of the discrepancy of the generated graphs from the test set (MMD RBF) does not change sensibly to the privacy parameter. On the other hand, the MMD RBF of the most competitive baseline BiGG does decrease with the privacy budget, as expected. This calls into question the effect of the Edge LDP guarantee on GenStat: it is possible that GenStat has a *fixed* privacy effect due to its immediate use of graph high-level graph statistics, rather than one that can be reasonably controlled with an Edge LDP technique. While it is nice that GenStat generates realistic graphs at all privacy budgets, this calls into question how much privacy is actually being preserved.

**Questions:**

Can the authors comment on my observations about Figure 4? Specifically, does the fact that GenStat maintains constant MMD RBF at all privacy budgets have implications about the true ability of GenStat to preserve privacy? Is there another empirical evaluation that could be done to show GenStat's response to a privacy budget?

**Limitations:**

Yes, the authors have adequately addressed limitations.

---

> ### Author Rebuttal · Authors · 2023-08-05
>
> We appreciate the reviewer's positive feedback, especially regarding their observation that "*The contribution itself is interesting and likely to impact graph learning beyond the specific application (graph generation) presented in the paper.*"
>
> **Questions.** Can the authors comment on my observations about Figure 4? Specifically, does the fact that GenStat maintains constant MMD RBF at all privacy budgets have implications about the true ability of GenStat to preserve privacy? Is there another empirical evaluation that could be done to show GenStat's response to a privacy budget?
>
> **Answer.** GenStat is indeed sensitive to the privacy budget (ε) in that the MMD-RBF is not constant at all privacy budgets. For example, in the Grid dataset (Figure 4.c), the MMD-RBF of the generated graph by GenStat decreases from ε=1 to ε=3 (larger ε; ε=0 indicates perfect privacy). Increasing the value of ε to 4, and larger, does not significantly improve the quality of generated graphs since GenStat has already reached its near-optimal performance.
>
> For the other two datasets, Mutag (Figure 4a) and PTC (Figure 4b), GenStat reaches its near-optimal performance even earlier with ε=1. Thus, for the given points with ε>1, the MMD-RBF curve looks flat. If we increase the resolution of the ε-axis to include ε=0.01, 0.1, and 0.5, we observe that the MMD-RBF of GenStat decreases as the privacy budget decreases; please see the following table, which uses the same experimental setting described in Section 4.3.
>
> Table: GenStat under ε-Edge LDP guarantee with different privacy budgets, in terms of the Random GNN-Based MMD RBF score. A lower score is better. A lower ε indicates a stricter privacy measure.
> | ε      | 0.01 | 0.1  | 0.5  | 1    | 2    | 3    | 4    |
> |--------|------|------|------|------|------|------|------|
> | Mutag  | 1.18 | 0.65 | 0.43 | 0.22 | 0.25 | 0.20 | 0.24 |
> | PTC    | 1.81 | 1.18 | 0.40 | 0.05 | 0.06 | 0.05  | 0.06 |
> | Grid   | 1.93 | 1.84 | 1.33 | 0.93 | 0.86 | 0.68 | 0.76 |
>
> We used the lower resolution with ε≥1 in the paper to focus on the comparison with the adjacency-based model (Bigg). However, we believe Figure 4 would benefit from an increased resolution. Thank you for raising this point.

---

> > ### Comment · Reviewer_92Y3 · 2023-08-19
> > **Concerns satisfactorily addressed.**
> >
> > Thanks for the response. It is indeed helpful to see the response of GenStat to the privacy budget, in the table that the authors newly provided in their response. As the authors suggested that they would do, it would be further helpful to extend the x-axis of Fig 4 accordingly and include the table numbers in the plot. I have increased my Soundess score and overall rating by one point each.

---

> > > ### Author Response · Authors · 2023-08-21
> > >
> > > We appreciate the reviewer's positive feedback and increasing the scores.

---

### Decision · Program_Chairs · 2023-09-21

**Decision:**

Accept (poster)

**Comment:**

This paper proposes privacy-aware graph generation from graph statistics. Its key idea is to use deterministic high-level graph statistics to represent a graph and decode a graph from the statistics. This preserves privacy in the sense that the generator does not observe individual nodes or edges for generating the graph.

The main concern of this problem was about the problem setting of privacy-aware generation, the lack of novelty compared to an existing work [Yoon et al., 2023], and whether the empirical evaluation properly shows the effectiveness of the proposed method under the considered scenario. After a thorough discussion between the reviewers, the reviewers came to an agreement that
- The considered problem-setting has real-world applications.
- The work is novel compared to [Yoon et al., 2023] since the existing work does not actually generate a graph and deals with the setting where a private, synthetic "doppelganger" of a single large graph is needed.

Upon reading the paper myself, I was a bit concerned that the graph being generated does not reflect the applications that actually require privacy-aware generation, e.g., why is privacy-aware generation necessary for Protein or Grid graphs? However, I think this is more about the lack of proper benchmarks fir graph generative models, so I down-weight this issue.

Overall, I recommend acceptance for this paper.